# Computational Modeling Analysis of Kinetics of Fumarate Reductase Activity and ROS Production during Reverse Electron Transfer in Mitochondrial Respiratory Complex II

**DOI:** 10.3390/ijms24098291

**Published:** 2023-05-05

**Authors:** Nikolay I. Markevich, Lubov N. Markevich

**Affiliations:** 1Institute of Theoretical and Experimental Biophysics of RAS, Pushchino, Moscow 142290, Russia; 2Institute of Cell Biophysics of RAS, Pushchino, Moscow 142290, Russia; lnmarkevich@mail.ru

**Keywords:** succinate dehydrogenase (SDH), a tunnel diode behavior, fumarate reduction, complex II, reactive oxygen species (ROS), computational model

## Abstract

Reverse electron transfer in mitochondrial complex II (CII) plays an important role in hypoxia/anoxia, in particular, in ischemia, when the blood supply to an organ is disrupted and oxygen is not available. A computational model of CII was developed in this work to facilitate the quantitative analysis of the kinetics of quinol-fumarate reduction as well as ROS production during reverse electron transfer in CII. The model consists of 20 ordinary differential equations and 7 moiety conservation equations. The parameter values were determined at which the kinetics of electron transfer in CII in both forward and reverse directions would be explained simultaneously. The possibility of the existence of the “tunnel diode” behavior in the reverse electron transfer in CII, where the driving force is QH_2_, was tested. It was found that any high concentrations of QH_2_ and fumarate are insufficient for the appearance of a tunnel effect. The results of computer modeling show that the maximum rate of succinate production cannot provide a high concentration of succinate in ischemia. Furthermore, computational modeling results predict a very low rate of ROS production, about 50 pmol/min/mg mitochondrial protein, which is considerably less than 1000 pmol/min/mg protein observed in CII in forward direction.

## 1. Introduction

Mitochondrial respiratory complex II (CII) plays an important role in the energy metabolism of cells because it links the Krebs cycle and the electron transport chain (ETC) by oxidizing succinate to fumarate and simultaneously supplying reduced ubiquinone to ETC. As was pointed earlier [1], the kinetics of electron transport in CII are characterized by strong nonlinearity, hysteresis in the forward direction [1], and rectification of current, the so-called a “tunnel diode” behavior, in the reverse direction of the electron flow [2].

Reverse electron transfer in CII plays an important role during hypoxia/anoxia, when there is not enough oxygen in the cell or none at all, as, for example, in ischemia.

It was shown [3] that complex I (CI) in anoxia reduces ubiquinone and reverses the flow of electrons in CII, resulting in the reduction of fumarate and the production of succinate. The production of succinate due to the reversal of the electron flow in CII during ischemia is of great importance since further oxidation of succinate can lead to injury at reperfusion [4]. However, it was noted that the strong accumulation of succinate may not be associated with the reverse of electrons in CII [5,6], in particular due to the effect of a tunnel diode. In our previous work [7], we analyzed the kinetic mechanisms of diode-like behavior of CII in the reverse quinol-fumarate reductase direction and the conditions when diode-like behavior is observed. The model quantitatively explained the experimentally observed effect of the tunnel diode, in particular, the values of threshold potentials under various conditions. However, we considered a simplified model of the reverse electron transfer to CII without taking into account the oxidation of the reduced ubiquinone QH_2_; that is, a model of only a water-soluble SDHA/SDHB subcomplex of CII was considered, which described an experimental procedure for the reduction of the [3Fe-4S] cluster by an electrode potential. In particular, it was shown that the tunnel effect is observed at a very high degree of cluster recovery (over 99%) at very high potential values exceeding a certain threshold value. Therefore, it was very important to understand whether the recovered ubiquinone QH_2_ could provide such a high degree of reduction of the [3Fe-4S] cluster, i.e., when the driving force of the reverse electron transfer is QH_2_, and not the electrode potential. Therefore, in this work, a general CII model was developed, including both electron transfer in a water-soluble SDHA/SDHB subcomplex and redox reactions in the ubiquinone-binding site. Using the general CII model, it was shown that the diode-like behavior effect in reverse electron transfer in CII is absent at any, even very high, values of QH_2_ and fumarate.

All these experimental data, as well as various hypotheses and assumptions, require a detailed theoretical analysis using mathematical models. In particular, the results of computer modeling show that the maximum rate of succinate production is not very high—30–40 times less than the rate of direct oxidation of succinate at the same values of the model parameters—and cannot provide a high concentration of succinate in ischemia. Thus, the results of this work are consistent with the assumption [6] that the main source of succinate in ischemia is the Krebs cycle upstream metabolites, but not the reverse electron transfer in the mitochondrial complex II.

Moreover, our computational modeling results predict very low values of the steady-state rate of ROS production, about 50 pmol/min/mg mitochondrial protein, which is considerably less than 1000 pmol/min/mg protein observed in CII in the forward direction [8]. However, Ref. [8] should be mentioned, in which it was shown that in the absence of respiratory chain inhibitors, model analysis revealed the [3Fe-4S] iron–sulfur cluster as the primary O_2_^·−^ source. In this case, taking into account the very high concentration of the cluster [3Fe-4S] in the reduced state, [3Fe-4S]^−^ at relatively small values of the QH_2_ and fumarate concentration, we should expect a high rate of ROS production by this cluster [3Fe-4S]^−^.

## 2. Results and Discussion

### 2.1. Effects of Changes in the Equilibrium Constants of Q and QH_2_ Binding to CII on the Kinetics of Electron Transfer in the Reverse Direction

The dependences of the electron transfer rate in the reverse direction—that is, the rate of succinate production during fumarate reduction—on the concentration of QH_2_ and fumarate at different values of the binding constants of QH_2_ and Q are shown in Figure 1. These are the equilibrium constants K_eq1_ and K_eq11_ of reactions 1 and 11, presented in Table 1. It was taken into account that K_eq11_ = 7.39/K_eq1_.

It is important to note that the values of the electron transfer rate constants between different redox centers as well as their redox potentials have been studied very well. Their values are presented in Table 1 with the corresponding citation of the literature. There are also literature data on the values of the dissociation/binding constants of succinate and fumarate with the decarboxylate binding center of complex II under different conditions. However, little is known about the values of the direct on and off constants binding/dissociation of ubiquinone, succinate, and fumarate with complex II. Therefore, we first assumed the values of these constants to be close to the maximum when analyzing the kinetics of electron transfer in CII. Therefore, when analyzing the stationary rate of fumarate reduction/succinate production during reverse electron transfer in CII, we first assumed that k_1_ = k_11_ = k_17_ = k_17a_ = k_19_ = k_20_ = k_21_ = k_21a_ = 100 s^−1^.

In addition, experimental data [9] and adjustable values of the binding/dissociation constants of oxidized and reduced ubiquinone, Q and QH_2_, obtained as a result of fitting a computational model of CII [8], show a difference of several orders of magnitude in the values of dissociation constants from 0.29 and 0.19 nM [8] up to 0.3 and 0.9 µM [9] for Q and QH_2_, respectively, which requires a careful analysis of the effect of this difference on the kinetics of electron transport in CII.

The results presented in Figure 1A,C show a non-monotonic dependence of the maximal steady-state rate of succinate production on the values of equilibrium constants K_eq1_ and K_eq11_. Figure 1A shows the computational modeling results using the values of the dissociation constants K_d_ for succinate and fumarate with CII experimentally obtained on bovine heart succinate–ubiquinone reductase [10]. The main feature of these dissociation constants is that they are different in the oxidized and reduced states of the enzyme. The K_d_ values are 10 and 240 µM for succinate and fumarate, respectively, in the oxidized state and 160 and 50 µM for succinate and fumarate in the reduced state of CII. In our case, this means that K_eq17_ = K_eq17a_ = 0.02 µM^−1^, K_eq20_ = 4.17 × 10^−3^ µM^−1^ for fumarate, and K_eq19_ = 10 µM, K_eq21_ = K_eq21a_ = 160 µM (see Table 2).

Figure 1A shows that after an initial increase in the maximal rate of succinate production from 237.8 up to 243 µM/s, with a simultaneous increase in the binding constant of K_eq1_ from 0.1 up to 10 µM^−1^ and a decrease in K_eq11_ from 73.9 to 0.739 µM, this rate decreases to 170 µM/s with a simultaneous increase in K_eq1_ up to 10^4^ µM^−1^ and a decrease in K_eq11_ to 7.39 × 10^−4^ µM. At the same time, the Michaelis constants for QH_2_ also behave nonmonotonically for each curve, first decreasing K_m_^QH2^ from 5.8 to 1.2 µM, with increasing K_eq1_ from 0.1 to 10 µM^−1^, and then increasing K_m_^QH2^ up to 20.2 µM with an increase in K_eq1_ up to 10^4^ µM^−1^ (Figure 1A). These are interesting results, meaning that an increase in the binding of QH_2_ and Q to CII leads first to an increase in the rate of succinate production and then to its decrease. It is clear that an increase in the binding of QH_2_ leads to an increase in the rate of succinate production, while an increase in the binding of Q leads to its decrease. Thus, with an increase in Q and QH_2_ binding—that is, a decrease in the dissociation of Q and QH_2_—the effect of an increase in QH_2_ binding prevails, first increasing the rate of succinate production, and then an increase in Q binding decreases the steady-state rate of succinate production.

The dependence of the stationary rate of succinate production on the fumarate concentration at different values of the equilibrium constants K_eq1_ and K_eq11_ is shown in Figure 1B. All the parameter values for Figure 1B are the same as in Figure 1A. It is very interesting that changes in K_eq1_ and K_eq11_ do not affect the dependence of the succinate production rate on the fumarate concentration at the total ubiquinone concentration equal to 500 µM. All curves, 1, 2, 3, and 4, completely coincide, although they are obtained at different values of K_eq1_ and K_eq11_, equal to 0.1, 0.5, 1 and 10 µM, respectively. All curves, 1, 2, 3, and 4, shown in Figure 1B, have the same K_m_^fum^ = 27 µM and the maximal rate V_max_ = 237.7 µM/s.

Thus, the computer simulation results presented in Figure 1A,B show that the model can account for the basic experimentally observed parameters such as the Michaelis constants for the rate of succinate production equal to 1.5 and 25 µM for QH_2_ and fumarate, respectively [24]. The computer simulation gives K_m_ = 27 µM for fumarate and K_m_ = 1.2 µM for QH_2_ at K_eq1_ = 10 µM^−1^ and K_eq11_ = 0.739 µM.

The computer-simulated value of catalytic constant for succinate production (fumarate reduction) by CII in this model—that is k_cat_ = V_max_/CII_tot_ ≈ 1 s^−1^, where CII_tot_ is the total concentration of CII in the inner membrane—is taken as 235 µM (see Table 1), close to the experimentally observed, for example, k_cat_ = 2 s^−1^.

Figure 1C,D differs from Figure 1A,B only in the values of the dissociation constants of succinate and fumarate with CII.

The K_d_ values for Figure 1C,D were taken from the work [8] and they have the following values: 355 and 1000 µM for succinate and fumarate, respectively, regardless of whether CII is in the oxidized or reduced state; that is, computations for Figure 1C,D were made at K_eq17_ = K_eq17a_ = K_eq20_ = 1 × 10^−3^ µM^−1^; K_eq19_ = K_eq21_ = K_eq21a_ = 355 µM.

Figure 1C, as well as Figure 1A, shows a non-monotonic dependence of the maximal steady-state rate of succinate production on QH_2_ concentration at different values of equilibrium constants K_eq1_ and K_eq11_. Figure 1C shows that after an initial increase in the maximal rate of succinate production from 105.9 up to 108.2 µM/s, with a simultaneous increase in the binding constant of K_eq1_ from 0.1 up to 10 µM^−1^ and a decrease in K_eq11_ from 73.9 to 0.739 µM, this rate decreases to 80 µM/s with a simultaneous increase in K_eq1_ up to 10^4^ µM^−1^ and a decrease in K_eq11_ to 7.39 × 10^−4^ µM. At the same time, the Michaelis constants for QH_2_ also behave nonmonotonically for each curve, first decreasing K_m_^QH2^ from 6.8 to 2 µM, with increasing K_eq1_ from 0.1 to 10 µM^−1^, and then increasing K_m_^QH2^ up to 17.2 µM, with an increase in K_eq1_ up to 10^4^ µM^−1^ (Figure 1C). Thus, the results presented in Figure 1C are very similar to those in Figure 1A. With an increase in binding—that is, a decrease in the dissociation of Q and QH_2_—the effect of an increase in QH_2_ binding prevails first, increasing the rate of succinate production, and then an increase in Q binding decreases the steady-state rate of succinate production.

Figure 1D is also very similar to Figure 1B. All the parameter values for Figure 1D are the same as in Figure 1C. In Figure 1D as well as in Figure 1B, changes in K_eq1_ and K_eq11_ do not affect the dependence of the succinate production rate on the fumarate concentration at the total ubiquinone concentration equal to 500 µM. All curves, 1, 2, 3, and 4, completely coincide, although they are obtained at different values of K_eq1_ and K_eq11_, equal to 0.1, 0.5, 1 and 10 µM, respectively. Curves 1, 2, 3, 4, shown in Figure 1B, have the same K_m_^fum^ = 86 µM and the maximal rate V_max_ = 105.7 µM/s.

Thus, Figure 1C,D shows higher values of K_m_ for QH_2_ and fumarate than in Figure 1A,B: K_m_^QH2^ = 2 µM at K_eq1_ = 10 µM^−1^ and K_m_^fum^ = 86 µM.

When analyzing the results presented in Figure 1, it is very important to note the absence of the diode-like behavior, which was observed experimentally during the reverse electron transfer in a soluble SDHA/SDHB subcomplex three decades ago [2]. The diode-like behavior or the tunnel diode effect in our case means a strong drop in the rate of fumarate reduction (succinate production) as the driving force is increased. Figure 1 shows the absence of the diode-like behavior. This is due to the insufficient driving force, that is, the insufficient thermodynamic gradient in the reverse transport of electrons in CII, which determines the rate of succinate production. Our earlier analysis of the tunneling effect in the reverse direction of electron transport in the AB subcomplex showed that the tunneling effect is observed only at a very high gradient of the thermodynamic potential, i.e., the electrode potential in relation to the mentioned experiment. In addition, we have shown that the tunnel effect is observed only at a very high degree (more than 99%) of oxidation of the [3Fe-4S] cluster. Figure 2 shows that neither QH_2_ nor fumarate can provide such a high degree of oxidation of the [3Fe-4S] cluster at the values of parameters considered in the model. The concentration of oxidized [3Fe-4S] cluster less than 200 µM at the total concentration of this cluster is equal to 235 µM; that is, the degree of oxidation of the [3Fe-4S] cluster is less than 85%. However, as shown in Figure 2C,D, the degree of [3Fe-4S] oxidation can approach 100% when the fumarate concentration approaches 0, i.e., equilibrium. This result is completely consistent with experimental observations [24] but does not affect the presence of the tunnel effect.

**Figure 1 ijms-24-08291-f001:**
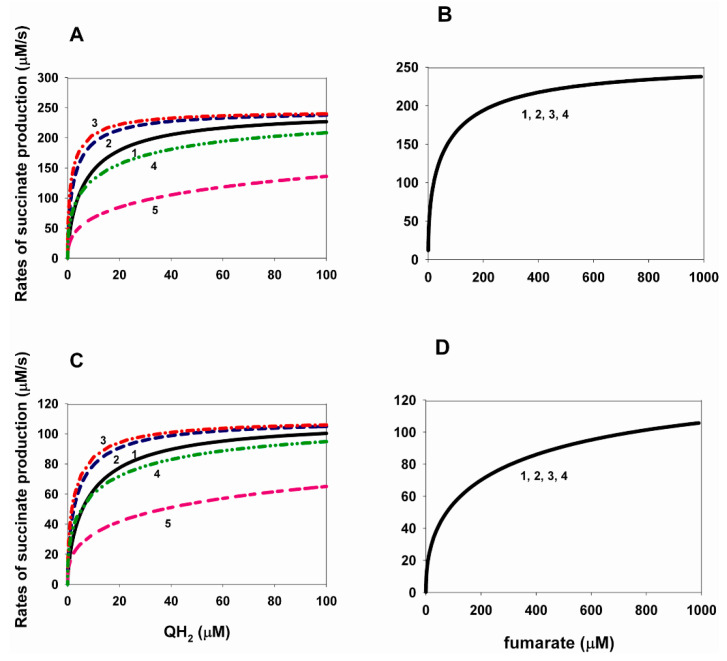
Computer-simulated dependence of the steady-state rate of succinate production during reverse electron transfer in SDH on the concentration of QH_2_ and fumarate at changes in the equilibrium constants of QH_2_ and Q binding to the Q-binding site. (**A**–**D**) Dependence of the rate of succinate production on QH_2_ (**A**,**C**) and fumarate (**B**,**D**) concentration at simultaneous changes in equilibrium constants K_eq1_ and K_eq11_ of the binding of QH2 and Q to the Q-binding site at the following values of kon and koff rate constants of binding/dissociation of Q, QH2, fumarate, and succinate to/from complex II: k_1_ = k_11_ = k_17_ = k_17a_ = k_19_ = k_20_ = k_21_ = k_21a_ = 100 s^−1^. The relation K_eq11_ = 7.39/K_eq1_ is taken into account. The values of equilibrium constants of succinate/fumarate binding to CII are taken from two different sources: [10] (**A**,**B**) K_eq17_ = K_eq17a_ = 0.02, K_eq20_ = 4.17 × 10^−3^ µM^−1^; K_eq19_ = 10, K_eq21_ = K_eq21a_ = 160 µM; and [8] (**C**,**D**) K_eq17_ = K_eq17a_ = K_eq20_ = 1 × 10^−3^ µM^−1^; K_eq19_ = K_eq21_ = K_eq21a_ = 355 µM. The values of the Michaelis constants, K_m_, are given for each curve. (**A**) The black solid curve (1) corresponds to K_eq1_ = 0.1 µM^−1^ and K_eq11_ = 73.9 µM (K_m_^QH2^ = 5.8 µM); blue dashed curve (2)—K_eq1_ = 0.5 µM^−1^ and K_eq11_ = 14.8 µM (K_m_^QH2^ = 2 µM); red dash-dot curve (3)—K_eq1_ = 10 µM^−1^ and K_eq11_ = 0.739 µM (K_m_^QH2^ = 1.2 µM); green dash-dot-dot curve (4)—K_eq1_ = 1 × 10^3^ µM^−1^ and K_eq11_ = 7.39 × 10^−3^ µM (K_m_^QH2^ = 6 µM); and pink short-long curve (5)—K_eq1_ = 1 × 10^4^ µM^−1^ and K_eq11_ = 7.39 × 10^−4^ µM (K_m_^QH2^ = 20.2 µM). (**B**) Simultaneous changes in the equilibrium constants K_eq1_ and K_eq11_ do not affect the dependence of the succinate production rate on the fumarate concentration at the total ubiquinone concentration equal to 500 µM. All curves, 1, 2, 3, and 4, completely coincide, although they are obtained at different values of K_eq1_ and K_eq11_, equal to 0.1, 0.5, 1 and 10 µM, respectively. Curves 1, 2, 3, 4, shown in B, have the same K_m_^fum^ = 27 µM. (**C**) Moreover, as in (**A**), the black solid curve (1) corresponds to K_eq1_ = 0.1 µM^−1^ and K_eq11_ = 73.9 µM (K_m_^QH2^ = 6.8 µM); the blue dashed curve (2)—K_eq1_ = 0.5 µM^−1^ and K_eq11_ = 14.8 µM (K_m_^QH2^ = 2.9 µM); the red dash-dot curve (3)—K_eq1_ = 10 µM^−1^ and K_eq11_ = 0.739 µM (K_m_^QH2^ = 2 µM); the green dash-dot-dot curve (4)—K_eq1_ = 1 × 10^3^ µM^−1^ and K_eq11_ = 7.39 × 10^−3^ µM (K_m_^QH2^ = 6 µM); and the pink short-long curve (5)—K_eq1_ = 1 × 10^4^ µM^−1^ and K_eq11_ = 7.39 × 10^−4^ µM (K_m_^QH2^ = 17.2 µM). (**D**) Simultaneous changes in the equilibrium constants K_eq1_ and K_eq11_ do not affect the dependence of the succinate production rate on the fumarate concentration at the total ubiquinone concentration equal to 500 µM. All curves, 1, 2, 3, and 4, also as in (**B**), completely coincide, although they are obtained at different values of K_eq1_ and K_eq11_, equal to 0.1, 0.5, 1 and 10 µM, respectively. Curves 1, 2, 3, 4, shown in (**D**), have the same K_m_^fum^ = 86 µM.

The rest of the model parameters are presented in Table 2. The succinate concentration is equal to 0.

### 2.2. Effects of Changes in the Equilibrium Constants of Q and QH_2_ Binding to Complex II on the Kinetics of Electron Transfer in the Forward Direction

For comparison, Figure 3 shows the dependencies of the electron transfer rate in the forward direction; that is, the rate of succinate oxidation on the concentration of Q and succinate at different values of binding constants of QH_2_ and Q; that is, equilibrium constants K_eq1_ and K_eq11_ of reactions 1 and 11. All the parameter values for Figure 3A,B and Figure 3C,D are the same as for Figure 1A,B and Figure 1C,D, respectively. It was taken into account that K_eq11_ = 7.39/K_eq1_.

The results presented in Figure 3 show a monotonic dependence of the maximal steady-state rate of succinate oxidation on the values of equilibrium constants K_eq1_ and K_eq11_. For the computational modeling results in Figure 3A,B the same parameter values as for Figure 1A,B were taken, that is, the dissociation constants K_d_ for succinate and fumarate with CII experimentally obtained on bovine heart succinate–ubiquinone reductase [10]. The following values of the equilibrium constant were taken: K_eq17_ = K_eq17a_ = 0.02 µM^−1^, K_eq20_ = 4.17 × 10^−3^ µM^−1^ for fumarate and K_eq19_ = 10 µM, K_eq21_ = K_eq21a_ = 160 µM (see Table 2).

The dependences of the steady-state rates of succinate oxidation on the oxidized Q concentration at different values of equilibrium constants K_eq1_ and K_eq11_ are presented in Figure 3A. Figure 3A shows a decrease in the maximal rate of succinate oxidation from 7297 to 470 µM/s with a simultaneous increase in the binding constant of K_eq1_ from 0.1 up to 50 µM^−1^ and a decrease in K_eq11_ from 73.9 to 0.147 µM, respectively. At the same time, the Michaelis constants for Q also behave monotonically for each curve, decreasing K_m_^Q^ from 64 to 0.34 µM with increasing K_eq1_ from 0.1 to 50 µM^−1^. It should be pointed out that the computer-simulated value K_m_^Q^ = 0.34 µM at K_eq1_ = 50 µM^−1^ and K_eq11_ = 0.147 µM is very close to the experimentally observed [24] value K_m_^Q^ = 0.3 µM.

Figure 3B shows the dependences of the electron transfer rate in the forward direction—that is, the rate of succinate oxidation—on the concentration of succinate at different values of equilibrium constants K_eq1_ and K_eq11_ of reactions 1 and 11 and at a constant total concentration of ubiquinone equal to 500 µM. As can be seen from Figure 3B, the maximum rate of succinate oxidation, V_max_, also monotonically decreases with increasing affinity of CII for Q and QH_2_, that is, an increase in K_eq1_ and a decrease in K_eq11_. V_max_ decreases from 6475 µM/s at K_eq1_ = 0.1 µM^−1^ and K_eq11_ = 73.9 µM with K_m_^suc^ = 2.9 µM to 2171 µM/s at K_eq1_ = 10 µM^−1^ and K_eq11_ = 0.739 µM with K_m_^suc^ = 0.75 µM. It should be pointed out that the computer-simulated values K_m_^suc^ at the used parameter values are significantly less than the experimentally observed [24] values K_m_^suc^, which are above 30 µM.

Figure 3C,D differs from Figure 3A,B only in the values of the dissociation constants of succinate and fumarate with CII. Moreover, all the parameter values for Figure 3C,D are the same as for Figure 1C,D.

The dependencies of the steady-state rates of succinate oxidation on the oxidized Q concentration at different values of the equilibrium constants K_eq1_ and K_eq11_ are presented in Figure 3C. Figure 3C, as well as Figure 3A, shows a decrease in the maximal rate of succinate oxidation from 10,020 to 475 µM/s with a simultaneous increase in the binding constant K_eq1_ from 0.1 up to 50 µM^−1^ and a decrease in K_eq11_ from 73.9 to 0.147 µM, respectively. At the same time, the Michaelis constants for Q also change monotonically for each curve, decreasing K_m_^Q^ from 69 to 0.33 µM with increasing K_eq1_ from 0.1 to 50 µM^−1^. It should be pointed that the computer-simulated value K_m_^Q^ = 0.33 µM at K_eq1_ = 50 µM^−1^ and K_eq11_ = 0.147 µM is also close to the experimentally observed [24] value K_m_^Q^ = 0.3 µM.

Figure 3D shows the dependencies of the rate of succinate oxidation on the concentration of succinate at different values of the equilibrium constants K_eq1_ and K_eq11_ of reactions 1 and 11 and at a constant total concentration of ubiquinone equal to 500 µM. As can be seen from Figure 3D, the maximum rate of succinate oxidation, V_max_, also monotonically decreases with increasing affinity of CII for Q and QH_2_, that is, an increase in K_eq1_ and a decrease in K_eq11_. V_max_ decreases from 7575 µM/s at K_eq1_ = 0.1 µM^−1^ and K_eq11_ = 73.9 µM with K_m_^suc^ = 62 µM to 254 µM/s at K_eq1_ = 100 µM^−1^ and K_eq11_ = 0.0739 µM with K_m_^suc^ = 2.05 µM. It should be pointed out that the computer-simulated values of K_m_^suc^ at parameter values used are close to the experimentally observed [24] values of K_m_^suc^, which are above 30 µM.

It should be noted that the ratio of the computer-simulated maximum electron transfer rates in the forward and reverse directions—that is, of succinate oxidation and fumarate reduction—are close to the experimentally observed values close to 30 [24].

It is also necessary to say a few words about the computer-simulated values of Michaelis constants for Q, QH_2_, succinate, and fumarate. Although they are close to the experimentally observed values, they are not at the same values of K_eq1_ (K_eq11_). With the values of direct binding constants (dissociation) used equal to 100 s^−1^, K_m_^Q^ values close to the experimentally observed 0.3 µM are obtained at very high values of K_eq1_ (low K_eq11_) close to 50 µM^−1^. However, with these values of K_eq1_, the values of K_m_^suc^ close to 1 µM are obtained, which is much less than the experimentally observed value above 30 µM. Therefore, it is necessary to change the direct constants of succinate binding/dissociation in order to obtain all model values of K_m_ close to experimental values at the same values of K_eq1_ and K_eq11_.

The rest of the model parameters are presented in Table 2. The fumarate concentration is equal to 0.

### 2.3. Effects of Changes in the Forward Constants of Q Dissociation and QH_2_ Binding to Complex II on the Kinetics of Electron Transfer in the Reverse and Forward Direction

As was noted in the previous Section 2.1 and Section 2.2, the approximation of the computer-simulated values of the Michaelis constants for the substrates of direct and reverse reactions in complex II to all experimentally observed values simultaneously cannot be obtained by changing only the equilibrium constants K_eq1_ and K_eq11_. The main problem is the very low obtained model values of K_m_^suc^ compared to the experimentally observed values above 30 mM, which are observed in the model with a low equilibrium constant of succinate binding to the decarboxylate binding site of CII (K_eq19_ = 10 µM [10]). Therefore, we made an attempt to obtain by computer simulation acceptable model values of K_m_ for all substrates in both forward and reverse directions of electron transfer in CII by changing other parameters of the model. The computer-simulated dependencies of the steady-state rates of succinate production and oxidation on the concentration of QH_2_, fumarate, and Q succinate during reverse and forward electron transfer in SDH, respectively, are shown in Figure 4. First of all, we used a mathematical model with adjustable values of the equilibrium constants of succinate and fumarate binding with CII: K_eq19_ = K_eq21_ = K_eq21a_ = 355 mM and K_eq17_ = K_eq17a_ = K_eq20_ = 10^−3^ µM^−1^ [8]. Then, we changed (increased) the values of the direct constants of the binding of QH_2_ with CII and the Q dissociation from the quinone-binding site; that is, k_1_ = 200 and k_11_ = 10,000 s^−1^ were taken instead of the initial values of k_1_ = k_1_ = 100 s^−1^. Computational simulations show that the model values of K_m_ for all curves shown in Figure 4 ((A) K_m_^QH2^ = 2.55 µM; (B) K_m_^fum^ = 72 µM; (C) K_m_^Q^ = 0.55 µM; and (D) K_m_^suc^ = 37 µM), although not ideal, are quite close to the experimentally observed values K_m_^QH2^ = 1.5, K_m_^fum^ = 25, K_m_^Q^ = 0.3, K_m_^suc^ = 30 µM [24].

In addition, Figure 5 shows the expected decrease in the steady-state rates of succinate production and oxidation on the concentration of the products of these reactions, succinate and fumarate, respectively.

### 2.4. Computational Analysis of the Rate of Reactive Oxygen Species (ROS) Production in Complex II during Reverse Electron Transfer

First of all, the dependence of the concentration of different ROS-producing redox centers of CII on the concentration of QH_2_ and fumarate as the substrates of the reverse reaction in CII is addressed. The steady-state dependencies of the concentration of FADH^∙^ and FADH_2_, as well as the cluster [3Fe-4S] in the reduced state and semiquinone anion (CII.Q**^.−^**), are presented in Figure 6A–C and Figure 7A–C. These redox centers were previously proposed [8,25,26] as the main ROS generators in CII. The main feature of these dependencies on both QH_2_ and fumarate concentrations is a very small degree of reduction of all ROS-forming redox centers with the exception of the cluster [3Fe-4S] (Figure 6B and Figure 7B). This means that the steady-state rate of ROS production in CII in the reverse direction should be very small if FADH^•^ and FADH_2_ are the major ROS producing redox centers FADH^•^ and FADH_2_, as shown earlier [25,26]. The QH_2_ and fumarate dependence of the stationary rates of ROS production is presented in Figure 6C and Figure 7C. The total stationary rate of ROS production was computed as the rate of H_2_O_2_ production and release from the mitochondrial matrix to cytosol, V_28_, which is equal to the summary rate of H_2_O_2_ production by FADH_2_, V_22_, and the dismutation of O_2_^.−^, V_27_, in the matrix at the steady state. The values of the catalytic constants of ROS formation by different redox centers in CII were taken from our previous computational model of CII [27], which accounted for the experimental data obtained on submitochondrial particles prepared from bovine [28] and rat heart [26] mitochondria upon inhibition of the Q-binding site by atpenin A5 and complex III (CIII) by myxothiazol, respectively.

All parameter values are the same as for Figure 4A. Fumarate concentration is equal to 1000 µM and succinate concentration is 0 for all curves.

All parameter values are the same as for Figure 4B. The total ubiquinone concentration is equal to 500 µM and succinate concentration is 0 for all curves.

## 3. Methods and Models

### 3.1. Kinetic Model of Reverse Electron Transfer in Mitochondrial Complex II

Reverse electron transfer in complex II is similar to direct electron transfer in quinol-fumarate reductase (QFR), which couples the two-electron oxidation quinol (QH_2_) to quinone (Q) to the two-electron reduction of fumarate (fum) to succinate (suc). In a simplified form, the overall catalytic reaction is as follows:QH_2_ + fum = Q + suc.(1)

A detailed kinetic scheme of reverse electron transfer and O_2_^.−^/H_2_O_2_ production underlying a mechanistic computational model of the respiratory complex II in the reverse direction (quinol-fumarate activity) is presented in Figure 8. Figure 8 presents both the mainstream reverse electron pathway from QH2 to fumarate (Figure 8A,B) and bypass reactions resulting in O_2_^.−^/H_2_O_2_ formation, which are marked in red in the kinetic scheme (Figure 8C). The scheme is supported by numerous literature data on electron transfer pathways between different redox centers of SDH [7,8,9,10]. This kinetic scheme includes the following electron carriers: (a) coenzyme Q; (b) heme *b* located in the SDHC/SDHD subcomplex; (c) three iron–sulfur clusters: [2Fe-2S], [4Fe-4S], and [3Fe-4S]) located in the SDHB subunit; (d) flavin adenine dinucleotide, FAD, located in the SDHA subunit.

The entire reaction network of electron transfer and O_2_^.−^/H_2_O_2_ production, which corresponds to this kinetic scheme in Figure 8 and consists of 27 reactions, is described in detail in Table 1. Additional reactions of release of hydrogen peroxide (H_2_O_2_) from the mitochondrial matrix to cytosol (reaction 28) and quinol (QH_2_) production in the mitochondrial inner membrane (reaction 29) complete the kinetic scheme of the model and are also presented in Table 1. Midpoint redox potentials, rate constants, and concentrations are taken from the experimental data (see Table 2 and references therein).

An initial part of the kinetic scheme of reverse electron transfer in CII, presented in Figure 8A, describes QH_2_ oxidation to Q in reactions 1–4 and 9–12. First, QH_2_ binds to the Q-binding site of CII in reaction 1 and releases one H^+^ in reaction 2, forming binding hydroquinone QH^−^ (CII.QH^−^). Then, QH^−^ donates the first electron to the cluster [3Fe-4S] in reaction 3 or to heme *b* in reaction 4, forming protonated semiquinone radical binding to CII, CII.QH^.^. After the second deprotonation of QH. and formation of Q^.−^ in reaction 9, binding semiquinone, CII.Q^.−^ donates the second electron to the cluster [3Fe-4S] in reaction 10 or to heme *b* in reaction 12, which results in the formation of CII.Q. The oxidized quinone Q then dissociates from the Q-binding site in reaction 11.

Thus, the cluster [3Fe-4S] accepting the first and second electrons either directly from the quinone (CII.QH^−^ or CII.Q^.−^) in reactions 3 and 10, respectively, or indirectly through heme *b* in reactions 5 and 13, transfers them further to the cluster [4Fe-4S] in reactions 6 and 14, and to the cluster [2Fe-2S] in reactions 7 and 15 (Figure 8A).

It should be noted that the initial part of the kinetic scheme of reverse electron transfer in CII presented in Figure 8A in reactions 1–4 and 9–12 is a somewhat simplified representation of the binding of the reduced ubiquinone QH_2_ to the Q site and its further oxidation to Q for two reasons. First, the existence of two quinone binding sites in CII is assumed. These are the proximal, Q_p_, and distal, Q_d_ quinone-binding sites. Although there is a general belief that only the Q_p_ site is involved in electron transfer. It was difficult to understand how the Q_d_ quinone could be involved in electron transfer because of the large distance (~27 Å) from its nearest redox center the quinone proximal to the [3Fe-4S] cluster (Q_p_). The function of the distal Q site, the Q_d_ site, is still unknown. Therefore, we assume in this work, following the authors of the generalized kinetic model of succinate oxidation by complex II [8], that only the Q_p_ site is involved in electron transfer. The second difficulty is that the oxidation/reduction of ubiquinone is not just the transfer of electrons and protons, but rather the movement of ubiquinone along the quinone-binding pocket and a change in the catalytic position. Our model is simplified in this case and combines the processes of electron and proton transfer with the mechanical movement of ubiquinone in the quinone-binding pocket. We believe that this simplification affects only the values of rate constants of the general oxidation/reduction of ubiquinone without affecting the overall kinetics of electron transfer in CII. However, in principle, taking into account the movement of a quinone in the quinone-binding pocket can affect the values of the electron and proton transfer rate constants.

Figure 8B is almost identical to Figure 10B in our previous work [7] and describes all the electron transfer paths in flavoprotein subunit A (SDHA), only with other reaction numbers. Therefore, in order not to repeat, all detailed explanations of the electron transfer paths in the SDHA subunit are given in Appendix A.

The main thing to note here is that the restrictions on the kinetic constants of the reactions along any thermodynamic cycle presented in Figure 8B must satisfy the so-called “detailed balance” relationships [25]. They are the following:K_eq8a_ ∙ K_eq16a_ ∙K_eq18_ ∙ K_eq19_ ∙ K_eq20_ = 1;K_eq16a_ ∙ K_eq17a_/(K_eq16_ ∙ K_eq17_) = 1; (2)K_eq8_ ∙ K_eq19_/(K_eq8b_ ∙ K_eq21a_) = 1;K_eq8b_ ∙K_eq16b_ ∙ K_eq17_ ∙ K_eq18_ ∙ K_eq21_ = 1.

Potential redox centers of ROS generation in the soluble subcomplex SDHA/SDHB are FADH^.^, FADH_2_, and [3Fe-4S] are presented in Figure 8C. These redox centers were previously proposed [7,8,26] as the main ROS generators in the subcomplex SDHA/SDHB of SDH.

The potential redox centers of ROS generation in the first electron transfer branch are FADH_2_, CII.Q**^.−^**, and [3Fe-4S], which are presented in (Figure 8C). FADH_2_ can generate either H_2_O_2_ in reaction 22 or superoxide in reaction 23. Semiquinone, CII.Q**^.^**, and the [3Fe-4S] cluster generate superoxide in reactions 25 and 26. The FADH. radical, which can be, in different states, an unoccupied dicarboxylate and occupied by fumarate or succinate, donates the second electron to the [2Fe-2S] cluster (reactions 14, 14a and 14b in Figure 1A) that transfers it to the protonated semiquinone CII.QH**^.^** in reactions 15–18 and 21. These reactions are followed by the formation and release of QH_2_ into the matrix (reactions 19–20 (Figure 8B)). In this branch of the second electron transfer, the potential redox centers of superoxide formation are the FADH^.^ radical (reaction 24) and [3Fe-4S] (reaction 26). In addition, intramitochondrial superoxide anion dismutation is represented by reaction 27.

### 3.2. Analysis of Parameter Values of the Kinetic Model of Reverse Electron Transfer in Mitochondrial Complex II

It should be pointed out that electron transfer between quinone, the cluster [3Fe-4S], and heme *b* form the thermodynamic cycle: Quinone ↔ [3Fe-4S] ↔ heme *b* ↔ Quinone (reactions 3, 5, and 4 for the first electron and reactions 10, 13, and 12 for the second electron transfer, respectively). In this case, we have to take into account “the principle of detail balancing” [29] for thermodynamic cycles, which requires the product of equilibrium constants along a cycle to be equal to 1. For the thermodynamic cycle Quinone ↔ [3Fe-4S] ↔ heme *b* ↔ Quinone, this means the following relations: K_eq4_·K_eq5_/K_eq3_ = 1 and K_eq12_·K_eq13_/K_eq10_ = 1 for the first and second electron transfer, respectively.

On the other hand, our [27] computational modeling analysis of electron transfer in CII with heme *b* in the forward SQR direction (succinate-CoQ reductase activity), i.e., including the thermodynamic cycle Quinone ↔ [3Fe-4S] ↔ heme *b* ↔ Quinone, showed that the value of the midpoint potential of the heme *b*, E_m_(*b*), does not affect the total rate of electron transfer in CII, that is, the rate of QH_2_ (Q) and suc (fum) formation in the steady state, although it does affect the rates of electron transfer inside the thermodynamic cycle, including heme *b*. This means that heme *b* can be excluded from the model if we analyze only the total rates of QH_2_ (Q) and suc (fum) formation in the steady state.

This result confirms suggestions advanced earlier [13,30] that heme *b* plays more of a structural role, stabilizing CII as a hetero-tetramer, than it is involved in catalysis in CII as an electron carrier.

That is why we can omit heme *b* and present catalytic reactions 1–3 and 9–11 of QH_2_ oxidation to Q in a simplified form as the overall catalytic reaction of QH2 oxidation:QH_2_ + 2·[3Fe-4S] = Q + 2·[3Fe-4S]^−^ + 2·H^+^. (3)

The equilibrium constant of this overall reaction (2) K_eq-overall_ can be presented as follows:K_eq-overall_ = exp (2 ∙ F ∙ (E_m,7_ ([3Fe-4S])-E_m,7_ (Q/QH_2_))/R ∙ T),(4)
where E_m,7_ ([3Fe-4S]) and E_m,7_ (Q/QH2) are the midpoint redox potentials of cluster [3Fe-4S] and quinone Q at pH = 7.

On the other hand, the summary equilibrium constant, K_eq-sum_, of the detailed reactions 1–3 and 9–11 of QH_2_ oxidation to Q, presented in Figure 8A, is the product of equilibrium constants of reactions 1–3 and 9–11:K_eq-sum_ = K_eq1_ · K_eq2_ · K_eq3_ · K_eq9_ · K_eq10_ · K_eq11._(5)

Here K_eq1_, K_eq2_, K_eq3_, K_eq9_, K_eq10_ and K_eq11_ are the equilibrium constants of the corresponding reactions 1–3 and 9–11.

According to the thermodynamic law of energy balance, these equilibrium constants K_eq-overall_ and K_eq-sum_ have to be equal each other:K_eq-overall_ = K_eq-sum._(6)

Let us consider this condition in more detail.

The values of all parameters in Equations (3) and (4) are presented in Table 2 with corresponding references except for E_m,7_ (Q/QH2).
K_eq1_ = 1.1 µM^−1^; K_eq2_ = 1.38 × 10^−5^ µM; K_eq3_ = 5.52 × 10^−3^; K_eq9_ = 4.9 µM; K_eq10_ = 6.67 × 10^3^; K_eq11_ = 0.3 µM;

So, for these values of equilibrium constants:K_eq-sum_ = K_eq1_ · K_eq2_ · K_eq3_ · K_eq9_ · K_eq10_ · K_eq11_ = 8.2 × 10^−4^ µM^2^.(7)

This value of K_eq-sum_ = 8.2 × 10^−4^ indicates a very strong shift in the reaction of oxidation of QH_2_ to Q in the reverse direction, that is, in the direction of reduction of Q to QH_2_, which is normal for succinate dehydrogenase.

It should be pointed out that the values of some of the constants mentioned above depend on the experimental conditions and vary greatly. For example, the value of the dissociation constant for Q and QH_2_ with the Q-binding site differs by 3–4 orders of magnitude in different literature data; that is, Q and QH_2_ bind preferentially to the reduced enzyme with the values of dissociation constant K_d_ of 0.3 and 0.9 µM, respectively, in the bovine enzyme [9]. At the same time, adjustable data for the computational model of SDH [8] predict that K_d_ = 0.29 nM and K_d_ = 0.19 nM for Q and QH_2_, respectively.

As to the value of the midpoint redox potential E_m,7_ (Q/QH_2_), it varies for ubiquinone in different conditions. For example, the standard redox potential of ubiquinone in submitochondrial particles from beef heart mitochondria E_m,7_ = 66 mV at pH = 7 and 25 °C. Very close values of E_m,7_ (Q/QH2) for ubiquinone approximately equal to 70 mV were obtained in phospholipid belayers [22].

At the same time, higher values of E_m_(Q/QH2) are noted in the literature, such as Em(Q/QH2) = 113 mV for *bos taurus* bovine mitochondria [9] and E_m,7_ (Q/QH_2_) = 90 mV for *E. coli* SQR [31].

In this work, we suggest that the value of E_m,7_(Q/QH2) for ubiquinone is equal to 90 mV.

Thus, taking into account Em([3Fe-4S]) = 60 mV, we can calculate the equilibrium constant for the reaction (3):K_eq-overall_ = exp [2F/RT * (Em([3Fe-4S]) − Em(Q/QH2))] = exp(2/25 * (60–90)) = exp(−60/25) = e^−2.4^ = 1/11.023 = 0.09.

Then, the ratio of the equilibrium constants of the overall and the sum of the detailed reactions significantly exceeds 1 and is equal to K_eq-overall_/K_eq-sum_ = 0.09/8.2 × 10^−4^ ≈ 110.

Thus, there is a big difference in the values of equilibrium constants for overall K_eq-overall_ and summarily detailed Keq-sum oxidation reactions of QH2 to Q at the values of various parameters taken from the literature. This is not very surprising, because as Rich also noted in his review [32], the properties of quinones presented in his review can be significantly altered by environmental changes and binding to specific sites of reaction on protein. In particular, it was shown [22] that the values of the midpoint redox potential E_m_(QH^.^/QH^−^) in phospholipid belayers are approximately 170 ± 30 mV, which is less than 190 mV observed in ethanol/water solutions [32]. However, the values of E_m,7_ (Q/QH2) in phospholipid belayers are approximately 72 ± 10 mV [22], which are equal to 70 mV observed in ethanol/water solutions [32].

In order for the ratio of these constants to be equal to 1, the constant K_eq-sum_ must be 110 times larger, which can be achieved by increasing one or more equilibrium constants of reactions 1–3 and 9–11.

First, we follow [22] and suggest that E_m_(QH^.^/QH^−^) = 150 mV instead of 190 mV observed in ethanol/water solutions [32]. In this case, a new adjusted value of K_eq3_ = exp[F ∙ (E_m_([3Fe-4S]) − E_m_(QH^.^/QH^−^))/R ∙ T] = exp((60–150)/25) = e^−3.6^ = 0.027. Thus, K_eq3_ increases by 0.027/5.52/10^−3^ = 4.9 times.

It should be taken into account that when E_m_(QH^.^/QH^−^) decreases from 190 to 150 mV, K_eq4_ increases from 3.06 × 10^−7^ (see Table 2) to 1.5 × 10^−6^.

Thus, when increasing K_eq3_ by 4.9 times, K_eq-sum_ also increases by 4.9 times up to 4.9 × 8.2 × 10^−4^ = 4.018 × 10^−3^. In this case, the ratio of the equilibrium constants of the overall and the sum of the detailed reactions still significantly exceeds 1 and is equal to K_eq-overall_/K_eq-sum_ = 0.09/4.018 × 10^−3^ = 22.4; or 110/4.9 = 22.4.

Adding to this value an increase in the constant K_eq1_ or K_eq11_ by 22.4 times (K_eq1_ equal to 24.64 instead of 1.1 µM^−1^ or K_eq11_ equal to 6.72 instead of 0.3 µM), we will obtain the necessary increase in K_eq-sum_ by 22.4 times and the ratio K_eq-overall_/K_eq-sum_ equal to 1.

In other words, the product of the constants K_eq1_ · K_eq11_ should be multiplied by 22.4; that is, instead of the initial value K_eq1_ · K_eq11_ = 1.1 × 0.3 = 0.33, the value should be 0.33 × 22.4 = 7.39. In this case, when varying the value of K_eq1_, we have to take into account that in order to comply with thermodynamic laws, the equality K_eq11_ = 7.39/K_eq1_ must hold.

We understand that these are approximate values of the considered equilibrium constants, which may differ from the real ones. Therefore, in this paper, we will consider the effect of changes in these parameters on the kinetics of electron transport in complex II.

At the same time, we will not fit the model parameter values to any particular series of experimental data; rather, we will undertake a general analysis of the effect of changes in model parameters on the experimentally observed kinetics of electron transport in the forward and reverse directions in complex II. However, we will mainly rely on the experimental study of fumarate activity of complex II performed by Prof. Vinogradov [24] and computational modeling of SDH performed in Prof. Basil’s laboratory [8].

### 3.3. Computational Model of Reverse Electron Transfer in CII. Mathematical Model

A computational model corresponding to the kinetic schemes in Figure 8 and Table 1 and Table 2 consists of 20 ordinary differential equations (ODE) and 7 moiety conservation equations. The models were implemented into DBSolve Optimum software available at website http://insysbio.ru. The differential equations of the model and Software Source Code are presented in Appendix A.

**Figure 8 ijms-24-08291-f008:**
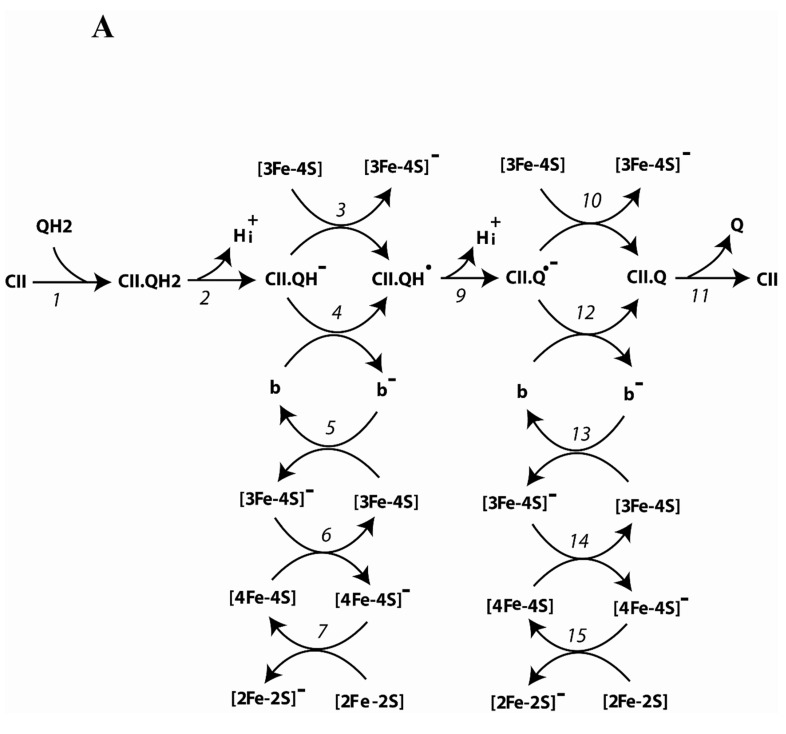
Kinetic schemes of reverse electron transfer and formation of superoxide anion, O_2_^.−^, and hydrogen peroxide, H_2_O_2_, in the respiratory complex II. (**A**) QH_2_ oxidation to Q in the Q-site and electron transfer through three [Fe–S] clusters located in the SDHB subunit. (**B**) Electron transfer and the interconversion of succinate and fumarate in the flavoprotein subunit SDHA. (**C**) Reactions of O_2_^.−^ and H_2_O_2_ formation in the complex II. The detailed reaction network is presented in Table 1 and Table 2.

## 4. Conclusions

A computational mechanistic model of electron transfer and formation of superoxide (O_2_^.−^) and hydrogen peroxide (H_2_O_2_) in the respiratory complex II (CII) of the inner mitochondrial membrane was developed to facilitate the quantitative analysis of the kinetics of quinol-fumarate reduction (QFR). The model corresponding to the kinetic schemes in Figure 8 and Table 1 and Table 2 consists of 20 ordinary differential equations and 7 moiety conservation equations, which describe the concentration of oxidized and reduced states of different redox centers and electron flows in CII.

In this work, several tasks were attempted. First, an attempt was made to determine the values of the parameters at which the kinetics of electron transport in CII in both forward and reverse directions would be explained simultaneously. For this purpose, the model parameters were determined at which the Michaelis constants of the substrates of direct and reverse reactions in CII were close to those experimentally observed. Computational simulations show that the model values of K_m_ for all curves, shown in Figure 4—((A) K_m_^QH2^ = 2.55 µM; (B) K_m_^fum^ = 72 µM; (C) K_m_^Q^ = 0.55 µM; (D) K_m_^suc^ = 37 µM)—although not ideal, are quite close to the experimentally observed values: K_m_^QH2^ = 1.5, K_m_^fum^ = 25, K_m_^Q^ = 0.3, K_m_^suc^ = 30 µM.

The second task was to determine the possibility of the existence of the “tunnel diode” behavior in the reverse electron transfer, discovered experimentally more than 30 years ago in the complete CII model. In our previous work [7], the tunnel effect in the reverse electron transfer in a water-soluble SDHA/SDHB subcomplex under the action of the electrode potential as the driving force of the reverse electron transfer was explained. It has been shown that the “tunnel diode” behavior occurs only with a very high reduction of cluster [3Fe-4S] (almost 99% of the cluster should be in the reduced state). In the present work, the possibility of the existence of the “tunnel diode” behavior in the reverse electron transfer in the full CII model, where the driving force is the reduced ubiquinone QH_2_, was tested. It was found that at high concentrations of QH_2_, the degree of cluster recovery does not exceed 80% and is insufficient for the appearance of a tunnel effect; that is, the dependence of the fumarate recovery rate (succinate production) on the concentration of QH_2_ is monotonic at any concentrations of QH_2_ and fumarate.

Moreover, the results of computer modeling showed that the maximum rate of succinate production is not very high, i.e., 30–40 times less than the rate of the direct oxidation of succinate at the same values of the model parameters, and cannot provide a high concentration of succinate in ischemia. Thus, the results of this work are consistent with the assumption [6] that the main source of succinate in ischemia is the Krebs cycle, but not the reverse electron transfer in the mitochondrial complex II.

Moreover, our computational modeling results predict very low values of the steady-state rate of ROS production, i.e., about 50 pmol/min/mg mitochondrial protein, which is considerably less than 1000 pmol/min/mg protein observed in CII in forward direction [26]. However, Ref. [26] should be mentioned, in which it was shown that in the absence of respiratory chain inhibitors, the model analysis revealed the [3Fe-4S] iron–sulfur cluster as the primary O_2_^.−^ source. In this case, taking into account the very high concentration of the cluster [3Fe-4S] in the reduced state, [3Fe-4S]^−^, at relatively small values of the QH_2_ and fumarate concentration, as shown in Figure 6B and Figure 7B, we should expect a high rate of ROS production by this cluster [3Fe-4S]^−^ of SDH in the reverse quinol-fumarate reductase direction in the absence of respiratory chain inhibitors. Thus, real changes in the rate of ROS production by CII in the reverse direction of SDH during hypoxia/anoxia—that is, during ischemia—depend on real experimental conditions.

## Figures and Tables

**Figure 2 ijms-24-08291-f002:**
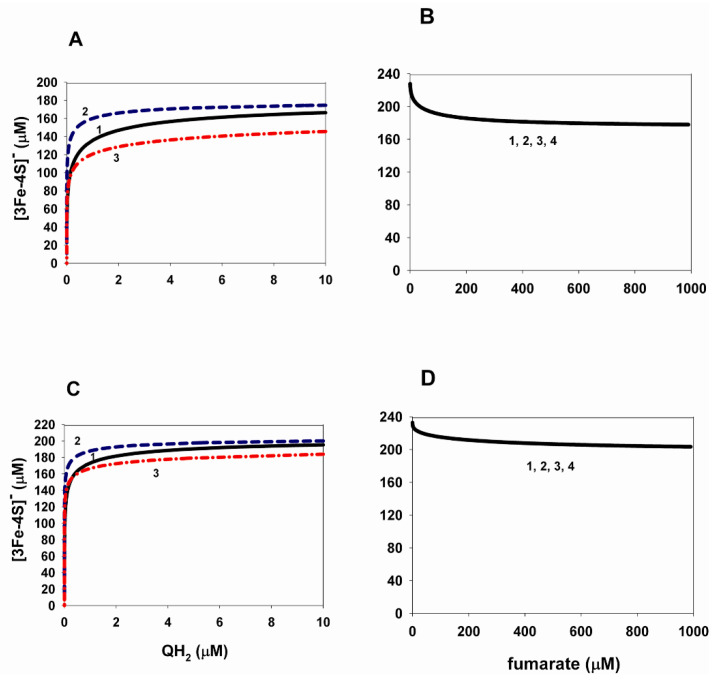
Computer-simulated dependence of the steady-state oxidized [3Fe-4S]^−^ concentration on the concentration of QH_2_ and fumarate at changes in the equilibrium constants of QH_2_ and Q binding to the Q-binding site during reverse electron transfer in SDH. (**A**–**D**) Dependence of the concentration of the oxidized cluster [3Fe-4S]^−^ on the QH_2_ (**A**,**C**) and fumarate (**B**,**D**) concentration in the steady-state at the same parameter values as in Figure 1A,B and Figure 1C,D, respectively. (**A**,**C**) Black solid curves (1) in both figures correspond to K_eq1_ = 0.1 µM^−1^ and K_eq11_ = 73.9 µM; blue dashed curves (2)—K_eq1_ = 10 µM^−1^ and K_eq11_ = 0.739 µM; red dash-dot curves (3)—K_eq1_ = 1 × 10^4^ µM^−1^ and K_eq11_ = 7.39 × 10^−4^ µM. (**B**,**D**) Simultaneous changes in the equilibrium constants K_eq1_ and K_eq11_ do not affect the dependence of the oxidized cluster [3Fe-4S]^−^ concentration on the fumarate concentration at the total ubiquinone concentration equal to 500 µM. Curves 1, 2, 3, and 4 in both (**B**,**D**) completely coincide, although they are obtained at different values of K_eq1_ and K_eq11_, equal to 0.1, 1, 10, 1 × 10^4^ µM^−1^ and 73.9, 7.39, 0.739, 7.39 × 10^−4^ µM, respectively.

**Figure 3 ijms-24-08291-f003:**
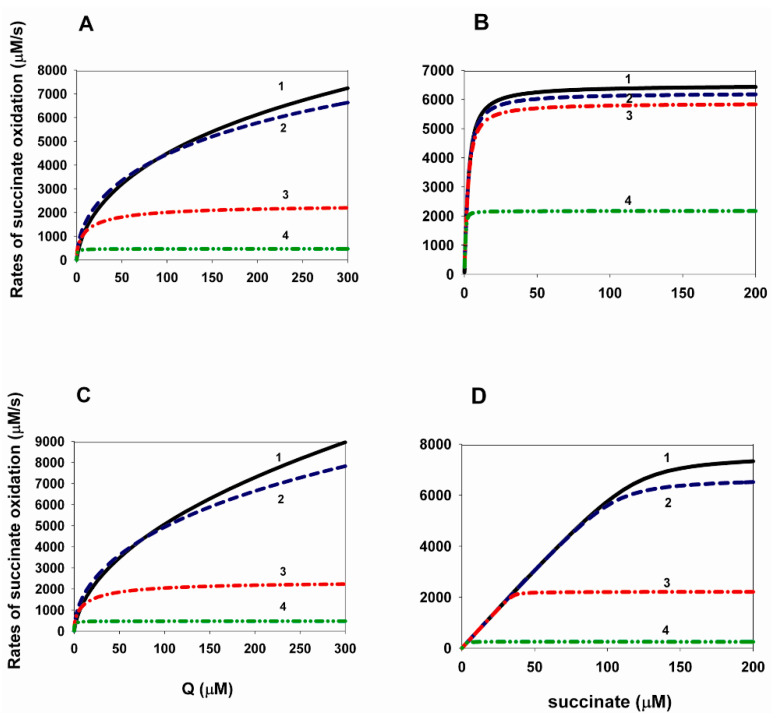
Computer-simulated dependence of the steady-state rate of succinate oxidation during forward electron transfer in SDH on the concentration of Q and succinate at changes in the equilibrium constants of QH_2_ and Q binding to the Q-binding site. (**A**–**D**) Dependence of the rate of succinate oxidation on Q (**A**,**C**) and succinate (**B**,**D**) concentration at simultaneous changes in equilibrium constants K_eq1_ and K_eq11_ of the binding of QH_2_ and Q to the Q-binding site at the following values of kon and koff rate constants of binding/dissociation of Q, QH_2_, fumarate, and succinate to/from complex II: k_1_ = k_11_ = k_17_ = k_17a_ = k_19_ = k_20_ = k_21_ = k_21a_ = 100 s^−1^. The relation K_eq11_ = 7.39/K_eq1_ is taken into account as well as in Figure 1 and Figure 2. The values of the Michaelis constants, K_m_, are given for each curve. (**A**) Black solid curve (1) corresponds to K_eq1_ = 0.1 µM^−1^ and K_eq11_ = 73.9 µM (K_m_^Q^ = 64 µM); blue dashed curve (2)—K_eq1_ = 1 µM^−1^ and K_eq11_ = 7.39 µM (K_m_^Q^ = 56 µM); red dash-dot curve (3)—K_eq1_ = 10 µM^−1^ and K_eq11_ = 0.739 µM (K_m_^Q^ = 8 µM); and green dash-dot-dot curve (4)—K_eq1_ = 50 µM^−1^ and K_eq11_ = 0.147 µM (K_m_^Q^ = 0.34 µM). (**B**) The total ubiquinone concentration is equal to 500 µM. Black solid curve (1) corresponds to K_eq1_ = 0.1 µM^−1^ and K_eq11_ = 73.9 µM (K_m_^suc^ =2.9 µM); blue dashed curve (2)—K_eq1_ = 0.5 µM^−1^ and K_eq11_ = 14.8 µM (K_m_^suc^ = 2.6 µM); red dash-dot curve (3)—K_eq1_ = 1 µM^−1^ and K_eq11_ = 7.39 µM (K_m_^suc^ = 2.5 µM); and green dash-dot-dot curve (4)—K_eq1_ = 10 µM^−1^ and K_eq11_ = 0.739 µM (K_m_^suc^ = 0.75 µM). All parameter values are the same as in Figure 1A,B. (**C**) Moreover, as in (**A**), the black solid curve (1) corresponds K_eq1_ = 0.1 µM^−1^ and K_eq11_ = 73.9 µM (K_m_^Q^ = 69 µM); blue dashed curve (2)—K_eq1_ = 1 µM^−1^ and K_eq11_ = 7.39 µM (K_m_^Q^ = 51 µM); red dash-dot curve (3)—K_eq1_ = 10 µM^−1^ and K_eq11_ = 0.739 µM (K_m_^Q^ = 7.5 µM); and green dash-dot-dot curve (4)—K_eq1_ = 50 µM^−1^ and K_eq11_ = 0.147 µM (K_m_^Q^ = 0.33 µM). (**D**) The total ubiquinone concentration is equal to 500 µM. The black solid curve (1) corresponds to K_eq1_ = 0.1 µM^−1^ and K_eq11_ = 73.9 µM (K_m_^suc^ = 62 µM); blue dashed curve (2)—K_eq1_ = 1 µM^−1^ and K_eq11_ = 7.39 µM (K_m_^suc^ = 54 µM); red dash-dot curve (3)—K_eq1_ = 10 µM^−1^ and K_eq11_ = 0.739 µM (K_m_^suc^ = 18 µM); and green dash-dot-dot curve (4)—K_eq1_ = 100 µM^−1^ and K_eq11_ = 0.0739 µM (K_m_^suc^ = 2.05 µM). All parameter values are the same as in Figure 1C,D.

**Figure 4 ijms-24-08291-f004:**
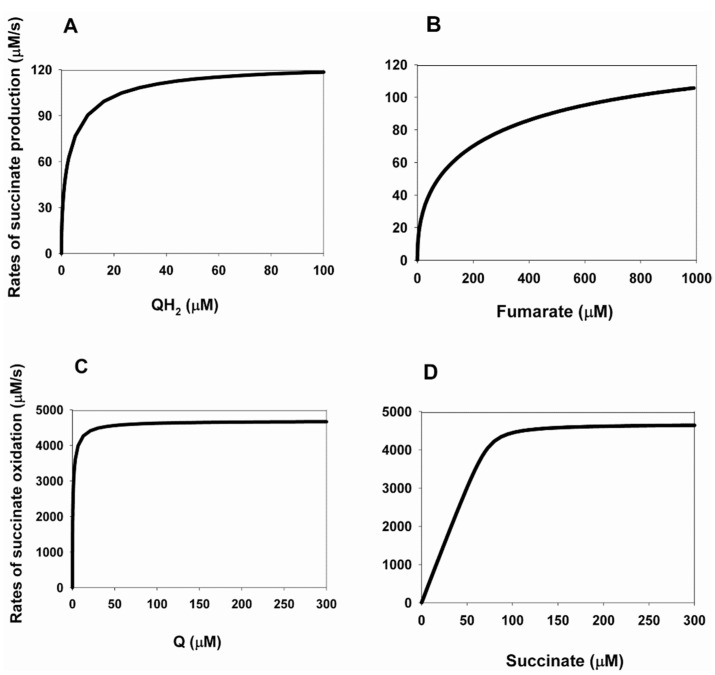
Computer-simulated dependence of the steady-state rate of succinate production and oxidation on the concentration of QH_2_, fumarate, and Q succinate during reverse and forward electron transfer in SDH, respectively. All parameter values for the computer simulation of fumarate reduction during reverse electron transfer (**A**,**B**) and succinate oxidation (**C**,**D**) are equal to each other and the same as for Figure 1C,D, excluding k_1_ = 200 and k_11_ = 10^4^ s^−1^ (Figure 4) instead of k_1_ = 100 and k_11_ = 100 s^−1^ (Figure 1, Figure 2 and Figure 3). (**A**–**D**) K_eq1_ = 10 µM^−1^ and K_eq11_ = 0.739 µM. Succinate concentration is taken to be 0 in (**A**,**B**) and fumarate concentration is 0 in (**C**,**D**). Fumarate (**A**) and succinate (**C**) concentrations are equal to 1000 µM. The Michaelis constants are the following: (**A**) K_m_^QH2^ = 2.55 µM; (**B**) K_m_^fum^ = 72 µM; (**C**) K_m_^Q^ = 0.55 µM; (**D**) K_m_^suc^ = 37 µM.

**Figure 5 ijms-24-08291-f005:**
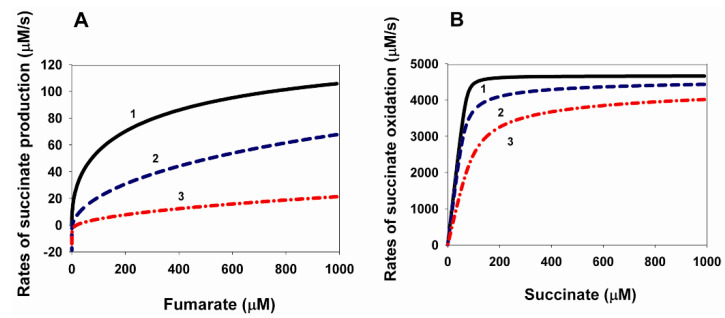
Computer-simulated dependence of the steady-state rate of succinate production and oxidation on the concentration of fumarate and succinate at different values of succinate and fumarate concentration, respectively. All parameter values for the computer simulation of fumarate reduction during reverse electron transfer (**A**) and succinate oxidation (**B**) are the same as for Figure 4B,D, respectively. (**A**) Succinate concentration is equal to 0, 10 and 100 µM for curves 1, 2, and 3, respectively. (**B**) Fumarate concentration is equal to 0, 100 and 1000 µM for curves 1, 2, and 3, respectively.

**Figure 6 ijms-24-08291-f006:**
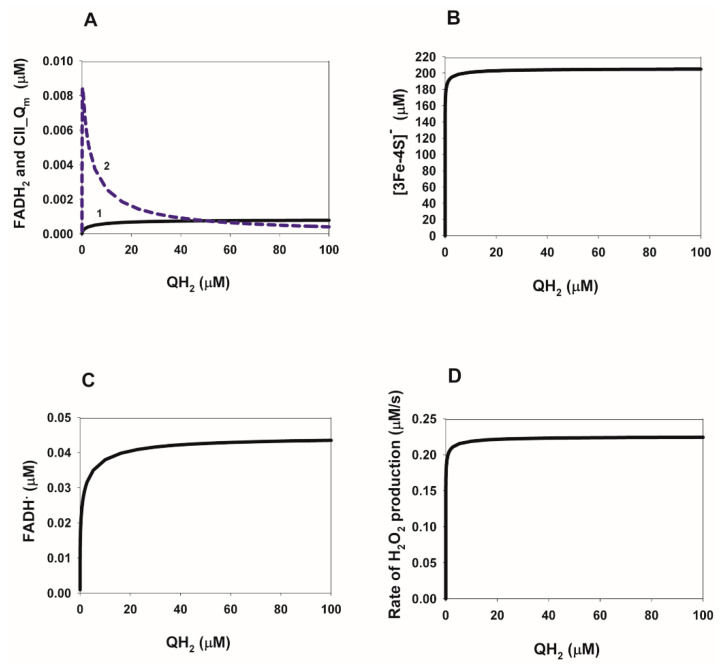
Computer-simulated dependence of the steady-state concentration of different redox ROS-generated redox centers, FADH, FADH_2_, cluster [3Fe-4S], in the reduced state and semiquinone anion (CII.Q^.−^) and the total rate of H_2_O_2_ production on the QH_2_ concentration during reverse electron transfer in SDH.

**Figure 7 ijms-24-08291-f007:**
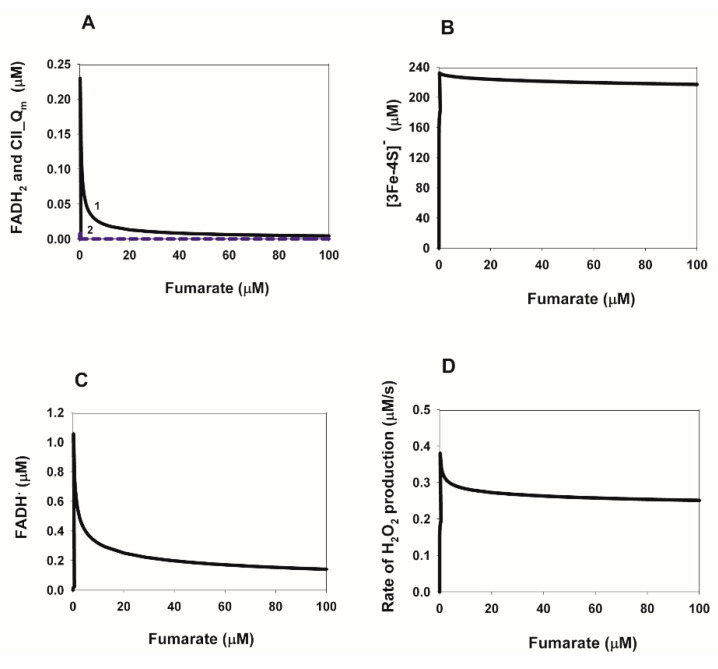
Computer-simulated dependence of the steady-state concentration of different redox ROS-generated redox centers, FADH, FADH_2_, cluster [3Fe-4S], in the reduced state and semiquinone anion (CII.Q^.−^) and the total rate of H_2_O_2_ production on the fumarate concentration during reverse electron transfer in SDH.

**Table 1 ijms-24-08291-t001:** Reactions and rate equations in reverse electron transfer in complex II.

No	Reaction	Rate Equation
**The First Electron Transfer**		
1	CII + QH2 = CII.QH2	V_1_ = k_1_ ∙ (CII ∙ QH2 − CII.QH2/K_eq1_)
2	CII.QH2 = CII.QH^−^ + H^+^	V_2_ = k_2_ ∙ (CII.QH2 − CII.QH^−^ ∙ H^+^/K_eq2_)
3	CII.QH^−^ + [3Fe-4S] = CII.QH^.^ + [3Fe-4S]^−^	V_3_ = k_3_ ∙ (CII.QH^−^ ∙ [3Fe-4S] − CII.QH^.^ ∙ [3Fe-4S]^−^/K_eq3_)
4	CII.QH^−^ + b = CII.QH^.^ + b^−^	V_4_ = k_4_ ∙ (CII.QH^−^ ∙ b − CII.QH^.^ ∙ b^−^/K_eq4_)
5	[3Fe-4S] + b^−^ = [3Fe-4S]^−^ + b	V_5_ = k_5_ ∙ ([3Fe-4S] ∙ b^−^ − [3Fe-4S]^−^ ∙ b/K_eq5_)
6	[4Fe-4S] + [3Fe-4S]^−^ = [4Fe-4S]^−^ + [3Fe-4S]	V_6_ = k_6_ ∙ ([4Fe-4S] ∙ [3Fe-4S]^−^ − [4Fe-4S]^−^ ∙ [3Fe-4S]/K_eq6_)
7	[2Fe-2S] + [4Fe-4S]^−^ = [2Fe-2S]^−^ + [4Fe-4S]	V_7_ = k_7_ ∙ ([2Fe-2S] ∙ [4Fe-4S]^−^ − [2Fe-2S]^−^ ∙ [4Fe-4S]/K_eq7_)
8	FAD + [2Fe-2S]^−^ + H^+^ = FADH^∙^ + [2Fe-2S]	V_8_ = k_8_ ∙ (FAD ∙ [2Fe-2S]^−^ ∙ H^+^ − FADH^∙^ ∙ [2Fe-2S]/K_eq8_)
8a	FAD.fum + [2Fe-2S]^−^ + H^+^ = FADH^∙^.fum+ [2Fe-2S]	V_8a_ = k_8a_ ∙ (FAD.fum ∙ [2Fe-2S]^−^ ∙ H^+^ − FADH^∙^.fum ∙ [2Fe-2S]/K_eq8a_)
8b	FAD.suc + [2Fe-2S]^−^ + H^+^ = FADH^∙^.suc + [2Fe-2S]	V_8b_ = k_8b_ ∙ (FAD.suc ∙ [2Fe-2S]^−^ ∙ H^+^ − FADH^∙^.suc ∙ [2Fe-2S]/K_eq8b_)
9	CII.QH^.^ = CII.Q^.−^ + H^+^	V_9_ = k_9_ ∙ (CII.QH^.^ − CII.Q^.−^ ∙ H^+^/K_eq9_)
**The second electron transfer**		
10	CII.Q^.−^ + [3Fe-4S] = CII.Q + [3Fe-4S]^−^	V_10_ = k_10_ ∙ (CII.Q^.−^ ∙ [3Fe-4S] − CII.Q ∙ [3Fe-4S]^−^/K_eq10_)
11	CII.Q = CII + Q	V_11_ = k_11_ ∙ (CII.Q − CII ∙ Q/K_eq11_)
12	CII.Q^.−^ + b = CII.Q + b^−^	V_12_ = k_12_ ∙ (CII.Q^.−^ ∙ b − CII.Q ∙ b^−^/K_eq12_)
13	[3Fe-4S] + b^−^ = [3Fe-4S]^−^ + b	V_13_ = k_13_ ∙ ([3Fe-4S] ∙ b^−^ − [3Fe-4S]^−^ ∙ b/K_eq13_)
14	[4Fe-4S] + [3Fe-4S]^−^ = [4Fe-4S]^−^ + [3Fe-4S]	V_14_ = k_14_ ∙ ([4Fe-4S] ∙ [3Fe-4S]^−^ − [4Fe-4S]^−^ ∙ [3Fe-4S]/K_eq14_)
15	[2Fe-2S] + [4Fe-4S]^−^ = [2Fe-2S]^−^ + [4Fe-4S]	V_15_ = k_15_ ∙ ([2Fe-2S] ∙ [4Fe-4S]^−^ − [2Fe-2S]^−^ ∙ [4Fe-4S]/K_eq15_)
16	FADH^∙^ + [2Fe-2S]^−^ + H^+^ = FADH2 + [2Fe-2S]	V_16_ = k_16_ ∙ (FADH^∙^ ∙ [2Fe-2S]^−^ ∙ H^+^ − FADH2 ∙ [2Fe-2S]/K_eq16_)
16a	FADH^∙^.fum + [2Fe-2S]^−^ + H^+^ = FADH2.fum + [2Fe-2S]	V_16a_ = k_16a_ ∙ (FADH^∙^.fum ∙ [2Fe-2S]^−^ ∙ H^+^ − FADH2.fum ∙ [2Fe-2S]/K_eq16a_)
16b	FADH^∙^.suc + [2Fe-2S]^−^ + H^+^ = FADH2.suc + [2Fe-2S]	V_16b_ = k_16b_ ∙ (FADH^∙^.suc ∙ [2Fe-2S]^−^ ∙ H^+^ − FADH2.suc ∙ [2Fe-2S]/K_eq16b_)
**Reduction of fumarate to succinate**		
17	FADH2 + fum = FADH2.fum	V_17_ = k_17_ ∙ (FADH2 ∙ fum − FADH2.fum/K_eq17_)
17a	FADH^∙^ + fum = FADH^∙^.fum	V_17a_ = k_17a_ ∙ (FADH^∙^ ∙ fum − FADH^∙^.fum/K_eq17a_)
18	FADH2.fum = FAD.suc	V_18_ = k_18_ ∙ (FADH2.fum − FAD.suc/K_eq18_)
19	FAD.suc = FAD + suc	V_19_ = k_19_ ∙ (FAD.suc − FAD ∙ suc/K_eq19_)
20	FAD + fum = FAD.fum	V_20_ = k_20_ ∙ (FAD ∙ fum − FAD.fum/K_eq20_)
21	FADH2.suc = suc + FADH2	V_21_ = k_21_ ∙ (FADH2.suc − FADH2 ∙ suc/K_eq21_)
21a	FADH^∙^.suc = suc + FADH^∙^	V_21a_ = k_21a_ ∙ (FADH^∙^.suc − FADH^∙^ ∙ suc/K_eq21a_)
**Hydrogen peroxide (H_2_O_2_) production by Complex II**		
22	FADH2 + O_2_ = FAD + H_2_O_2_	V_22_ = k_22_ ∙ (FADH2 ∙ O_2_ − FAD ∙ H_2_O_2_/K_eq22_)
**Superoxide anion (O_2_^.−^) production by Complex II**		
23	FADH2 + O_2_ = FADH^∙^ + O_2_^.−^ + H^+^	V_23_ = k_23_ ∙ (FADH2 ∙ O_2_ − FADH^∙^ ∙ O_2_^.−^ ∙ H^+^/K_eq23_)
24	FADH^∙^ + O_2_ = FAD + O_2_^.−^ + H^+^	V_24_ = k_24_ ∙ (FADH^∙^ ∙ O_2_ − FAD ∙ O_2_^.−^ ∙ H^+^/K_eq24_)
25	[3Fe-4S]^−^ + O_2_ = [3Fe-4S] + O_2_^.−^	V_25_ = k_25_ ∙ ([3Fe-4S]^−^ ∙ O_2_ − [3Fe-4S] ∙ O_2_^.−^/K_eq25_)
26	CII.Q^.−^ + O_2_ = CII.Q + O_2_^.−^	V_26_ = k_26_ ∙ (CII.Q^.−^ ∙ O_2_ − CII.Q ∙ O_2_^.−^/K_eq26_)
**Accompanying reactions in the matrix and inner membrane**		
**Superoxide anion dismutation in the mitochondrial matrix**		
27	2O_2_^.−^ + 2H^+^ → O_2_ + H_2_O_2_	V_27_ = V_max27_ ∙ O_2_^.−^/(K_m27_ + O_2_^.−^)
**Release of hydrogen peroxide (H_2_O_2_) from the mitochondrial matrix to cytosol**		
28	H_2_O_2_ →	V_28_ = k_28_ ∙ H_2_O_2_
**Ubiquinol (QH2) production in the mitochondrial inner membrane**		
29	Q + 2H^+^ → QH2	V_29_ = k_29_ ∙ Q

**Table 2 ijms-24-08291-t002:** Parameter values for the model.

Reaction No	Midpoint Potential E_m_ = E, (mV)	Equilibrium Constant K_eq_	k_forward_	Other Parameters	Reference
**The First Electron Transfer**					
1		1.1 µM^−1^	100 µM^−1^·s^−1^		[9] ^b^
2		1.38 × 10^−5^ µM ^e^	100 s^−1^	pH = 7	[11] ^b^
3	E([3Fe-4S]) = 60 E(QH^.^/QH^−^) = 190	5.52 × 10^−3^	39.7 µM^−1^·s^−1^	pH = 7.4	[12] ^a^ [13] ^c^
4	E(b) = −185 E(QH^.^/QH^−^) = 190	3.06 × 10^−7^	8824 µM^−1^·s^−1^		[14] ^a^ [13] ^c^
5	E([3Fe-4S]) = 60 E(b) = −185	1.8 × 10^4^	1.52 × 10^7^ µM^−1^·s^−1^	pH = 7.4 pH = 7	[12,14] ^a^ [13] ^c^
6	E([4Fe-4S]) = −260 E([3Fe-4S]) = 60	2.78 × 10^−6^	1 × 10^3^ µM^−1^·s^−1^	pH = 7.4	[15] ^a^ [12] ^a^
7	E([2Fe-2S]) = 0 E([4Fe-4S]) = −260	3.29 × 10^4^	1 × 10^6^ µM^−1^·s^−1^	pH = 7.4	[15] ^a^
8	E(FAD/FADH^∙^) = −127 E([2Fe-2S]) = 0	0.006	1 × 10^3^ µM^−2^·s^−1^	pH = 7 pH = 7.4	[15,16] ^a^
8a	E(FAD/FADH^∙^) = −127 E([2Fe-2S]) = 0	0.03 ^g^ µM^−1^	1 × 10^3^ µM^−2^·s^−1^	pH = 7 pH = 7.4	[15,16] ^a^
8b	E(FAD/FADH^∙^) = −127 E([2Fe-2S]) = 0	3.75 × 10^−4 g^ µM^−1^	1 × 10^3^ µM^−2^·s^−1^	pH = 7 pH = 7.4	[15,16] ^a^
9		4.9 µM ^e^	1 × 10^6^ s^−1^	pH = 7	[12] ^b^
**The second electron transfer**					
10	E([3Fe-4S]) = 60 E(Q/Q^.−^) = −163	6.67 × 10^3^	4.8 × 10^7^ µM^−1^·s^−1^	pH = 7.4 pH = 7	[12,17] ^a^ [13] ^c^
11		0.3 µM	100 s^−1^		[9] ^b^
12	E(b) = −185 E(Q/Q^.−^) = −163	0.37	8824 µM^−1^·s^−1^	pH = 7	[17,14] ^a^ [13] ^c^
13	E([3Fe-4S]) = 60 E(b) = −185	1.8 × 10^4^	1.52 × 10^7^ µM^−1^·s^−1^		[12,14] ^a^ [13] ^c^
14	E([4Fe-4S]) = −260 E([3Fe-4S]) = 60	2.78 × 10^−6^	1 × 10^3^ µM^−1^·s^−1^	pH = 7.4	[15] ^a^ [12] ^a^
15	E([2Fe-2S]) = 0 E([4Fe-4S]) = −260	3.29 × 10^4^	3.3 × 10^8^ µM^−1^·s^−1^	pH = 7.4	[15] ^a^
16	E(FADH^∙^/FADH2) = −30 E([2Fe-2S]) = 0	0.289 µM^−1^	1 × 10^3^ µM^−2^·s^−1^	pH = 7 pH = 7.4	[16] ^a^ [15] ^a^
16a	E(FADH^∙^/FADH2) = −30 E([2Fe-2S]) = 0	0.289 ^g^ µM^−1^	1 × 10^3^ µM^−2^·s^−1^	pH = 7 pH = 7.4	[16] ^a^ [15] ^a^
16b	E(FADH^∙^/FADH2) = −30 E([2Fe-2S]) = 0	0.47 ^g^ µM^−1^	1 × 10^3^ µM^−2^·s^−1^	pH = 7 pH = 7.4	[16] ^a^ [15] ^a^
**Reduction of fumarate to succinate**					
17		0.02 µM^−1^	100 s^−1^		[10] ^b^
17a		0.02 µM^−1^	100 s^−1^		[10] ^b^
18		2778 ^d^	2.78 × 10^6^ s^−1^		
19		10 µM	100 s^−1^		[10] ^b^
20		4.17 × 10^−3^ µM^−1^	100 s^−1^		[10] ^b^
21		160 µM	100 s^−1^		[10] ^b^
21a		160 µM	100 s^−1^		[10] ^b^
**Hydrogen peroxide (H_2_O_2_) production by complex II**					
22	E(O_2_/H_2_O_2_)= 690 E(FAD/FADH2) = −79	5.2 × 10^26^	0.01 µM^−1^·s^−1^	pH = 7	[16] ^a^
**Superoxide anion (O_2_^.−^) production by complex II**					
23	E(O_2_/O_2_^.−^) = −160 E(FADH^∙^/FADH2) = −31	6 × 10^−3^	0.01 µM^−1^·s^−1^	pH = 7	[18] ^a^ [16] ^a^
24	E(O_2_/O_2_^.−^) = −160 E(FAD/FADH^∙^) = −127	0.267	0.1 µM^−1^·s^−1^	pH = 7 pH = 7.4	[18] ^a^ [16] ^a^
25	E(O_2_/O_2_^.−^) = −160 E([3Fe-4S]) = 60	1.5 × 10^−4^	1 × 10^−3^ µM^−1^·s^−1^	pH = 7 pH = 7.4	[18] ^a^ [12] ^a^
26	E(O_2_/O_2_^.−^) = −160 E(Q/Q^.−^) = −163	1	0.05 µM^−1^·s^−1^		[17,18] ^a^
**Accompanying reactions in the matrix and inner membrane**					
**Superoxide anion dismutation in the mitochondrial matrix**					
27				V_max27_ = 5.6 × 10^4^ µM·s^−1 f^ K_m27_ = 50 µM	[19] ^d^
**Efflux of hydrogen peroxide (H_2_O_2_) from the mitochondrial matrix to cytoplasm**					
28			30 s^−1^		[20] ^c^
**Ubiquinol (QH2) production in the mitochondrial inner membrane**					
29			1 s^−1^		

^a^ The reference for the midpoint redox potential E_M_. ^b^ The reference for the equilibrium constant K_eq_. ^c^ The reference for the rate constant of direct reaction k_forward_. ^d^ The K_eq18_ value used is calculated from the relation K_eq17_ ∙ K_eq18_ ∙ K_eq19_ = exp (2 ∙ F ∙ (E (fum/suc)–E (FAD/FADH2))/R ∙ T) = 555.6 according to the thermodynamic cycle, where midpoint redox potentials E (FAD/FADH2) = −79 mV (pH 7.0) and E (fum/suc) = 0 mV (pH 7.0) [20], respectively, and F, R, and T have the usual meaning. So, K_eq18_ = 555.6/K_eq19_/K_eq17_ = 555.6/10/0.02 = 2778. ^e^ The equilibrium constant K_eq2_ used here corresponds to pK_a_ = 10.86 for the ubiquinone pair QH2./QH^−^ in aqueous solution, taken from [2]. So, K_eq2_ = 10^−10.86^ M = 1.38 × 10^−5^ µM. By analogy, equilibrium constant K_eq9_ corresponds to pK_a_ = 5.31 for the ubiquinone pair QH**^.^**^.^/Q^.–^ [11]. K_eq9_ = 10^−5.31^ M = 4.9 μM. ^f^ The value used was taken from [19], which was calculated from experimental data on Mn–SOD activity in mitochondria of cardiac cells [21]. ^g^ Relations between equilibrium constants according to four thermodynamic cycles in the kinetic scheme are taken from system (3.2) in Section 3. Methods and models. K_eq16a_ = K_eq16_ · K_eq17_/K_eq17a_ = 0.289 × 0.02/0.02 = 0.289; K_eq8a_ = 1/K_eq20_/K_eq16a_/K_eq18_/K_eq19_ = 1/4.17 × 10^−03^/0.289/2778/10 = 0.02987; K_eq8b_= K_eq19_ · K_eq8_/K_eq21a_ = 10 × 0.006/160 = 3.75 × 10^−4^; K_eq16b_ = 1/(K_eq17_ ∙ K_eq18_ ∙ K_eq8b_ ∙ K_eq21_) = 1/0.02/2778/2.4 × 10^−4^/160 = 0.47. *Conserved moieties (in µM).* The pool of electron carriers was calculated in our previous paper [22]. According to [23], the content of complex II in cardiac mitochondria is 0.209 nmol complex II/mg of mitochondrial protein. Translation of whole membrane concentration expressed in nmol/mg mit.prot. to local protein concentration expressed in μM is presented in [22]. We have shown earlier [22] that 1 nmol/mg of protein corresponds to 273 μM when normalized to the mitochondrial volume (V_mit_). If the concentration is normalized to the inner mitochondrial membrane volume (V_imb_), it should be additionally taken into account that the ratio W_imb_ = V_imb_/V_mit_ = 0.24 [22]. Therefore, the 0.209 nmol complex II/mg of mitochondrial protein corresponds approximately to 235 μM if it is recalculated to the concentration in the inner MM (0.209 × 273/0.24 = 237). So, we suggested in the present study that the total concentration of all redox centers localized in complex II—that is, [FAD], [2Fe–2S], [4Fe–4S], and [3Fe–4S]—is equal to 235 μM. The total concentration of coenzyme Q in the inner membrane was taken to be 4541 μM as in [22].

## Data Availability

Not applicable.

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
