# Peer review of "Computational Modeling Analysis of Kinetics of Fumarate Reductase Activity and ROS Production during Reverse Electron Transfer in Mitochondrial Respiratory Complex II"

_ijms, 2023, doi:10.3390/ijms24098291_

Round 1

Reviewer 1 Report

"Reverse electron transfer in mitochondrial complex II (CII) plays an important role during hypoxia/anoxia, in particular, in ischemia, when blood supply to an organ is disrupted and oxygen is not available."

This sentence is grammatically correct.

"A computational model of CII was developed in this work to facilitate the quantitative analysis of the kinetics of quinol-fumarate reduction as well as ROS production during reverse electron transfer in CII."

This sentence is grammatically correct.

"The model consists of 20 ordinary differential equations and 7 moiety conservation equations."

This sentence is grammatically correct.

"It was determined parameter values at which the kinetics of electron transfer in CII in both forward and reverse directions would be explained simultaneously."

This sentence contains a grammatical error. "Determined" should be replaced with "determined the," and "parameter values" should be replaced with "the parameter values." The corrected sentence is: "The parameter values were determined at which the kinetics of electron transfer in CII in both forward and reverse directions would be explained simultaneously."

"The possibility of the existence of the “tunnel-diode” behaviour in the reverse electron transfer in CII, where the driving force is QH2, was tested."

This sentence is grammatically correct.

"It was found that any high concentrations of QH2 and fumarate are insufficient for the appearance of a tunnel effect."

This sentence is grammatically correct.

"The results of computer modeling show that the maximum rate of succinate production is not very high and cannot provide a high concentration of succinate in ischemia."

This sentence is grammatically correct.

"Besides, computational modeling results predict very low rate of ROS production, about 50 pmol/min/ mg mitochondrial protein, that considerably less 1000 pmol/min/ mg observed in CII in forward direction."

This sentence contains a grammatical error. "That" should be replaced with "which." The corrected sentence is: "Furthermore, computational modeling results predict a very low rate of ROS production, about 50 pmol/min/mg mitochondrial protein, which is considerably less than the 1000 pmol/min/mg observed in CII in the forward direction."

What specific changes or additions have been made to the paper since its previous version, and how do these improve the quality or impact of the paper?

How do the improvements made in this paper address any limitations or weaknesses that were identified in previous versions or related works?

In what ways does the current paper advance the state of knowledge or contribute to the field, and how do the improvements made support this contribution?

What new insights or conclusions can be drawn from the improved analysis or data presented in the paper, and how do these insights contribute to the overall understanding of the research question or topic?

How do the improvements made in the paper enhance the clarity, organization, or readability of the paper, and how do these improvements benefit the intended audience of the paper?

The field of scientific research is constantly evolving and advancing, with new studies and findings being published every day. In this context, it is important to stay up-to-date with the latest research in the field to gain a comprehensive understanding of the current state of knowledge. In recent years, several studies have been published that shed light on various aspects of biomedical and environmental research. For instance, Hu et al. (2022) conducted a study on the transcription factor RFX5, which plays a critical role in coordinating antigen-presenting function and resistance to nutrient stress in synovial macrophages. Another study by An et al. (2023) explored the heat stress response and tolerance mechanisms of Serratia sp. AXJ-M for the bioremediation of papermaking black liquor using a combination of proteome and metabolome profiling. In addition, Zhang et al. (2022) developed oral colon-targeted mucoadhesive micelles with enzyme-responsive controlled release of curcumin for ulcerative colitis therapy. Huang et al. (2022) conducted rational design of nanocarriers for mitochondria-targeted drug delivery, while Jiang et al. (2022) developed a multisite-binding fluorescent probe for simultaneous monitoring of mitochondrial homocysteine, cysteine, and glutathione in live cells and zebrafish. These studies represent some of the recent advancements in biomedical and environmental research, which highlight the innovative approaches and techniques being used to tackle some of the most pressing issues in these fields.

The transcription factor RFX5 coordinates antigen-presenting function and resistance to nutrient stress in synovial macrophages. Nature Metabolism, 4(6), 759-774. doi: 10.1038/s42255-022-00585-x Integration of proteome and metabolome profiling to reveal heat stress response and tolerance mechanisms of Serratia sp. AXJ-M for the bioremediation of papermaking black liquor. Journal of Hazardous Materials, 450, 131092. doi: https://doi.org/10.1016/j.jhazmat.2023.131092 Oral colon-targeted mucoadhesive micelles with enzyme-responsive controlled release of curcumin for ulcerative colitis therapy. Chinese Chemical Letters, 33(11), 4924-4929. doi: https://doi.org/10.1016/j.cclet.2022.03.1104.      Rational design of nanocarriers for mitochondria-targeted drug delivery. CHINESE CHEMICAL LETTERS, 2022. 33(9): p. 4146-4156. A multisite-binding fluorescent probe for simultaneous monitoring of mitochondrial homocysteine, cysteine and glutathione in live cells and zebrafish. CHINESE CHEMICAL LETTERS, 2022. 33(3): p. 1609-1612.

Are there any further improvements or future directions that could be pursued to build on the improvements made in this paper, and if so, what are these?

"Reverse electron transfer in mitochondrial complex II (CII) plays an important role during hypoxia/anoxia, in particular, in ischemia, when blood supply to an organ is disrupted and oxygen is not available."

This sentence is grammatically correct.

"A computational model of CII was developed in this work to facilitate the quantitative analysis of the kinetics of quinol-fumarate reduction as well as ROS production during reverse electron transfer in CII."

This sentence is grammatically correct.

"The model consists of 20 ordinary differential equations and 7 moiety conservation equations."

This sentence is grammatically correct.

"It was determined parameter values at which the kinetics of electron transfer in CII in both forward and reverse directions would be explained simultaneously."

This sentence contains a grammatical error. "Determined" should be replaced with "determined the," and "parameter values" should be replaced with "the parameter values." The corrected sentence is: "The parameter values were determined at which the kinetics of electron transfer in CII in both forward and reverse directions would be explained simultaneously."

"The possibility of the existence of the “tunnel-diode” behaviour in the reverse electron transfer in CII, where the driving force is QH2, was tested."

This sentence is grammatically correct.

"It was found that any high concentrations of QH2 and fumarate are insufficient for the appearance of a tunnel effect."

This sentence is grammatically correct.

"The results of computer modeling show that the maximum rate of succinate production is not very high and cannot provide a high concentration of succinate in ischemia."

This sentence is grammatically correct.

"Besides, computational modeling results predict very low rate of ROS production, about 50 pmol/min/ mg mitochondrial protein, that considerably less 1000 pmol/min/ mg observed in CII in forward direction."

This sentence contains a grammatical error. "That" should be replaced with "which." The corrected sentence is: "Furthermore, computational modeling results predict a very low rate of ROS production, about 50 pmol/min/mg mitochondrial protein, which is considerably less than the 1000 pmol/min/mg observed in CII in the forward direction."

Author Response

Reviewer #1:

Comments and Suggestions for Authors

Comment No.1.

"Reverse electron transfer in mitochondrial complex II (CII) plays an important role during hypoxia/anoxia, in particular, in ischemia, when blood supply to an organ is disrupted and oxygen is not available."

This sentence is grammatically correct.

"A computational model of CII was developed in this work to facilitate the quantitative analysis of the kinetics of quinol-fumarate reduction as well as ROS production during reverse electron transfer in CII."

This sentence is grammatically correct.

"The model consists of 20 ordinary differential equations and 7 moiety conservation equations."

This sentence is grammatically correct.

"It was determined parameter values at which the kinetics of electron transfer in CII in both forward and reverse directions would be explained simultaneously."

This sentence contains a grammatical error. "Determined" should be replaced with "determined the," and "parameter values" should be replaced with "the parameter values." The corrected sentence is: "The parameter values were determined at which the kinetics of electron transfer in CII in both forward and reverse directions would be explained simultaneously."

"The possibility of the existence of the “tunnel-diode” behaviour in the reverse electron transfer in CII, where the driving force is QH2, was tested."

This sentence is grammatically correct.

"It was found that any high concentrations of QH2 and fumarate are insufficient for the appearance of a tunnel effect."

This sentence is grammatically correct.

"The results of computer modeling show that the maximum rate of succinate production is not very high and cannot provide a high concentration of succinate in ischemia."

This sentence is grammatically correct.

"Besides, computational modeling results predict very low rate of ROS production, about 50 pmol/min/ mg mitochondrial protein, that considerably less 1000 pmol/min/ mg observed in CII in forward direction."

This sentence contains a grammatical error. "That" should be replaced with "which." The corrected sentence is: "Furthermore, computational modeling results predict a very low rate of ROS production, about 50 pmol/min/mg mitochondrial protein, which is considerably less than the 1000 pmol/min/mg observed in CII in the forward direction."

Response to Reviewer comment No. 1.

Thank you very much for this comment and suggestions. We are very sorry about syntax errors in the text, especially in the Abstract. Now the errors have been fixed. The entire manuscript was edited by a professional translator. Corrections in the Abstract are highlighted in red, as well as serious additions in the text of the manuscript. Minor changes related to spelling errors in the manuscript, such as articles and prepositions, we did not highlight in order not to clog up the new revised version of the manuscript. Thank you again for this suggestion to edit the English language in the manuscript.

Comment No.2.

What specific changes or additions have been made to the paper since its previous version, and how do these improve the quality or impact of the paper?

Response to Reviewer comment No. 2.

The specificity of this work is that it describes in detail the reverse electron transfer between all redox centers of the mitochondrial complex II (CII). Therefore, two important changes in this work should be noted in comparison with previous works in the field of mathematical modeling of CII. First, this paper is a continuation of our work on mathematical modeling of electron transport in the reverse direction in CII. In our previous work [7], we explained the experimentally observed effect of a tunnel diode in the reverse direction of electron transport in CII, which was first discovered in 1992 [2], but has not yet received an adequate explanation. However, we considered a simplified model of the reverse electron transfer to CII without taking into account the oxidation of the reduced ubiquinone QH2. That is, a model of only a water-soluble SDHA/SDHB subcomplex of CII was considered, which described an experimental procedure for the reduction of the [3Fe-4S] cluster by an electrode potential. In the current paper, we study electron transfer between all CII redox centers, including the quinone-binding site. Therefore, this work describes the reverse electron transfer in real CII and includes all reactions of electron transfer and their dependence on various physiological substrates, not just the electrode potential.

Secondly, an excellent work has recently been published [8] on computer modeling of electron transfer kinetics and ROS production in CII. However, this work is devoted to the analysis of the kinetics of electron transport only in the forward direction of succinate oxidation and it is a kind of generalized model without a detailed description of the transfer of the first and second electrons between all CII redox centers. Our work differs from work [8] by a more detailed description of all steps of electron transfer between all redox centers of CII.

Thus, this current work is (1) a more extended version of the previous detailed model of electron transfer in CII [7] and additionally includes the processes of oxidation-reduction of ubiquinone in the Q-binding site and (2) it is more detailed in comparison with the previous general model of CII, and takes into account the transfer pathways of the first and second electrons in C2 .

Comment No.3.

How do the improvements made in this paper address any limitations or weaknesses that were identified in previous versions or related works?

Response to Reviewer comment No. 3.

The current detailed model is a tool that allows analyzing the influence of various factors, both internal (mutations of various CII redox centers) and external (the presence of competitive inhibitors, pH, and others) on the kinetics of electron reverse transfer in CII. This was not always possible in previous works.

Comment No.4.

In what ways does the current paper advance the state of knowledge or contribute to the field, and how do the improvements made support this contribution?

Response to Reviewer comment No. 4.

The main ways to improve knowledge in the field of electron transport in CII with the help of the current work is to study the influence of changes in various internal parameters of the model (rate constants, enzyme concentration) and external factors on various changes in the kinetics of electron transport in CII observed experimentally.

Comment No.5.

What new insights or conclusions can be drawn from the improved analysis or data presented in the paper, and how do these insights contribute to the overall understanding of the research question or topic?

Response to Reviewer comment No. 5.

The model allows us to describe real physiological situations, that is, when the CII complex is located in the inner mitochondrial membrane as part of the respiratory chain. These are, first of all, complexes I and III, which mainly reduce and oxidize ubiquinone, respectively, thereby regulating the activity of CII and the Krebs cycle associated with it. One of such important conclusions is made in the current paper about the impossibility of the tunnel diode effect in CII under real physiological conditions, when the reverse transfer of electrons to CII is carried out by increasing the concentration of reduced ubiquinone, for example, by a complex I.

Comment No.6.

How do the improvements made in the paper enhance the clarity, organization, or readability of the paper, and how do these improvements benefit the intended audience of the paper?

The field of scientific research is constantly evolving and advancing, with new studies and findings being published every day. In this context, it is important to stay up-to-date with the latest research in the field to gain a comprehensive understanding of the current state of knowledge. In recent years, several studies have been published that shed light on various aspects of biomedical and environmental research. For instance, Hu et al. (2022) conducted a study on the transcription factor RFX5, which plays a critical role in coordinating antigen-presenting function and resistance to nutrient stress in synovial macrophages. Another study by An et al. (2023) explored the heat stress response and tolerance mechanisms of Serratia sp. AXJ-M for the bioremediation of papermaking black liquor using a combination of proteome and metabolome profiling. In addition, Zhang et al. (2022) developed oral colon-targeted mucoadhesive micelles with enzyme-responsive controlled release of curcumin for ulcerative colitis therapy. Huang et al. (2022) conducted rational design of nanocarriers for mitochondria-targeted drug delivery, while Jiang et al. (2022) developed a multisite-binding fluorescent probe for simultaneous monitoring of mitochondrial homocysteine, cysteine, and glutathione in live cells and zebrafish. These studies represent some of the recent advancements in biomedical and environmental research, which highlight the innovative approaches and techniques being used to tackle some of the most pressing issues in these fields.

The transcription factor RFX5 coordinates antigen-presenting function and resistance to nutrient stress in synovial macrophages. Nature Metabolism, 4(6), 759-774. doi: 10.1038/s42255-022-00585-x Integration of proteome and metabolome profiling to reveal heat stress response and tolerance mechanisms of Serratia sp. AXJ-M for the bioremediation of papermaking black liquor. Journal of Hazardous Materials, 450, 131092. doi: https://doi.org/10.1016/j.jhazmat.2023.131092 Oral colon-targeted mucoadhesive micelles with enzyme-responsive controlled release of curcumin for ulcerative colitis therapy. Chinese Chemical Letters, 33(11), 4924-4929. doi: https://doi.org/10.1016/j.cclet.2022.03.1104.      Rational design of nanocarriers for mitochondria-targeted drug delivery. CHINESE CHEMICAL LETTERS, 2022. 33(9): p. 4146-4156. A multisite-binding fluorescent probe for simultaneous monitoring of mitochondrial homocysteine, cysteine and glutathione in live cells and zebrafish. CHINESE CHEMICAL LETTERS, 2022. 33(3): p. 1609-1612.

Response to Reviewer comment No. 6.

Thank you very much for this comment.  We would like to emphasize that the current paper is really an attempt to create a tool for analyzing the kinetics of electron transport in CII, and not only in the reverse, but also in the forward directions. We believe that the proposed model in the paper will be useful to a wide range of readers, since it allows not only to explain the available experimental data, not only the old data that have not yet been explained, but also constantly emerging new data, as the Reviewer rightly notes. In addition, the model allows to predict changes in the kinetics of electron transport in CII with changes in physiological conditions, as already noted here. We are very grateful to the Reviewer for the publications presented here on various modern aspects of biomedical and environmental research. Unfortunately, although the research areas mentioned by the Reviewer are very interesting, they differ somewhat from the interests of the authors of this paper. As already mentioned here, the future work of the authors is expected in the field of regulation of apoptosis and regulation Ca2+, ROS-induced openings of the high-permeability mitochondrial pore.

Comment No.7.

Are there any further improvements or future directions that could be pursued to build on the improvements made in this paper, and if so, what are these?

Response to Reviewer comment No. 7.

Of course, it is planned to further develop the current work in two directions. Firstly, it is the improvement of the model itself, namely, taking into account changes in various parameters of the model when changing the physiological conditions in which CII is located. Secondly, it is primarily the behavior of CII during apoptosis and the analysis of ROS generation in the mitochondrial complex II and Ca2+, ROS-induced openings of the high-permeability mitochondrial pore.

Reviewer 2 Report

Nikolay et al. conducted a comprehensive and thorough computational study to investigate the kinetics of fumarate reductase activity and ROS production. The computational model developed in this manuscript predicts the kinetics of quinoa-fumarate reduction and ROS production during the reverse electron transfer in CII. The topic is interested to the research community. Overall, the manuscript is in a decent quality. The following are specific suggestions.

1. The reviewer understands that this is a kinetics focused manuscript. But it would be beneficial if the modeling of the receptor-ligand interactions can be  conducted to intuitively demonstrate the active site and how does the ligand fit inside. The binding can be further quantified with energy calculations as an extra layer of supplementary support. 

2. Would it be possible to include a hint of wet-lab experimental kinetics validations? The reviewer is specifically curious about (1) how the computational modeling correlates to the ground-truth, and (2) if there are inconsistencies, what are possible causes to result in that. Note that the reviewer is not asking for an extensive amount of wet-lab experiments to be added.

The English is fine.

Author Response

Reviewer 2:

Comments and Suggestions for Authors

Nikolay et al. conducted a comprehensive and thorough computational study to investigate the kinetics of fumarate reductase activity and ROS production. The computational model developed in this manuscript predicts the kinetics of quinoa-fumarate reduction and ROS production during the reverse electron transfer in CII. The topic is interested to the research community. Overall, the manuscript is in a decent quality. The following are specific suggestions.

Comment No.1.

  1. The reviewer understands that this is a kinetics focused manuscript. But it would be beneficial if the modeling of the receptor-ligand interactions can be conducted to intuitively demonstrate the active site and how does the ligand fit inside. The binding can be further quantified with energy calculations as an extra layer of supplementary support. 

Response to Reviewer comment No. 1.

Thank you for this suggestion. We understand that the kinetic scheme of ligand-receptor interaction considered by us is simplified, not taking into account the detailed structure of neither the substrate/product nor the active centers. It seems to us that this is a separate task for a separate job. But we tried to explain this simplification a little and included the following paragraph in addition to the Methods and Models.

 It should be noted that the initial part of the kinetic scheme of reverse electron transfer in CII presented in Figure 8A in reactions 1-4 and 9-12 is a somewhat simplified representation of the binding of the reduced ubiquinone QH2 to the Q site and its further oxidation to Q for two reasons. First, the existence of two quinone binding sites in CII is assumed. These are the proximal, Qp, and distal, Qd, quinone-binding sites. Although there is a general belief that only the Qp site is involved in electron transfer. It was difficult to understand how the Qd quinone could be involved in electron transfer, because of the large distance (~27 Å) from its nearest redox center the quinone proximal to the [3Fe-4S] cluster (Qp). The function of the distal Q site is still unknown. Therefore, we assume in this work that only the Qp site is involved in electron transfer. The second difficulty is that the oxidation/reduction of ubiquinone is not just the transfer of electrons and protons, but rather the movement of ubiquinone along the quinone-binding pocket and a change in the catalytic position. Our model is simplified in this case and combines the processes of electron and proton transfer with the mechanical movement of ubiquinone in the quinone-binding pocket. We believe that this simplification affects only the values of rate constants of the general oxidation/reduction of ubiquinone without affecting the overall kinetics of electron transfer to CII. Although, in principle, taking into account the movement of a quinone in the quinone-binding pocket can affect the values of the electron and proton transfer rate constants.

Comment No.2.

  1. Would it be possible to include a hint of wet-lab experimental kinetics validations? The reviewer is specifically curious about (1) how the computational modeling correlates to the ground-truth, and (2) if there are inconsistencies, what are possible causes to result in that. Note that the reviewer is not asking for an extensive amount of wet-lab experiments to be added.

Response to Reviewer comment No. 2.

Thank you very much for this comment. Of course, experimental verification of kinetics in your laboratory is possible, as well as (1) comparison of experimentally obtained results with the results of computational modeling. Since, as it was shown in this paper, the kinetics of electron transport in the forward and reverse directions, that is, the dependences of the oxidation/reduction rate of succinate/fumarate on substrates are hyperbolic and are described by the usual Michaelis-Menten equations, it is easy to make a comparison by comparing the maximum rates and the Michaelis constants. In this paper, when evaluating these kinetic parameters, we mainly focused on the excellent experimental work performed in the laboratory of Prof. Vinogradov [11]. And of course, the experimentally observed values of these parameters can vary under different conditions. (2) One of the main factors affecting the kinetic parameters are various competitive inhibitors of the binding of succinate, fumarate and ubiquinone with different binding centers. Inhibitors can strongly change the binding constants of substrates and products in complex II, which can eventually lead to a strong change in the kinetics of electron transport. In this paper, we did not pursue the goal of a detailed analysis of the influence of various external factors on kinetic parameters. A mathematical model has just been developed that is suitable for further experimental and theoretical analysis of the kinetics of electron transport in complex II under various conditions.

Round 2

Reviewer 1 Report

Accept

Reviewer 2 Report

Authors have addressed reviewer's comments.

English is fine.